# Linear Contextual Bandits with Adversarial Corruptions

## Abstract

We study the linear contextual bandits problem in the presence of adversarial corruption, where the interaction between the player and a possibly infinite decision set is contaminated by an adversary that can corrupt the reward up to a corruption level $C$ measured by the sum of the largest alteration on rewards in each round. We present a variance-aware algorithm that is adaptive to the level of adversarial contamination $C$. The key algorithmic design includes (1) a multi-level partition scheme of the observed data, (2) a cascade of confidence sets that are adaptive to the level of the corruption, and (3) a variance-aware confidence set construction that can take advantage of low-variance reward. We further prove that the regret of the proposed algorithm is $\widetilde{O}(C^2 d\sqrt{\sum_{t=1}^{T} \sigma_t^2} + C^2\sqrt{dT} + CR\sqrt{dT})$, where $d$ is the dimension of context vectors, $T$ is the number of rounds, $R$ is the range of noise and $\sigma_t^2, t = 1\ldots, T$ are the variances of instantaneous reward. We also prove a gap-dependent regret bound for the proposed algorithm, which is instance-dependent and thus leads to better performance on good practical instances. To the best of our knowledge, this is the first variance-aware corruption robust algorithm for contextual bandits.

## 1 Introduction

Multi-armed bandits algorithms are widely applied in online advertising (Li et al., 2010), clinical trials (Villar et al., 2015), recommendation system (Deshpande and Montanari, 2012) and many other real-world tasks. In the model of multi-armed bandits, the algorithm needs to decide which action (or arm) to take (or pull) at each round and receive a reward for the chosen action. In the stochastic setting, the reward is subject to a fixed but unknown distribution for each action. In reality, however, these rewards can easily be "corrupted" by some malicious users. A typical example is click fraud (Lykouris et al., 2018), where botnets simulate the legitimate users clicking on an ad to fool the recommendation systems. This motivates the studies of the bandits algorithms that are robust to adversarial corruptions.

For example, Lykouris et al. (2018) introduced a bandit model in which an adversary could corrupt the stochastic reward generated by an arm pull. They proposed an algorithm and show that the regret of this "middle ground" scenario degrades smoothly with the amount of corruption injected by the adversary. Gupta et al. (2019) proposed an alternative algorithm which gives a significant improvement in regret.

While the algorithms that are robust to the corruptions have been studied in the setting of multi-armed bandits in a number of prior works, they are still understudied in the setting of linear contextual bandits. The linear contextual bandits problem can be regarded as an extension of the multi-armed bandit problem to linear optimization, in order to tackle an unfixed and possibly infinite set of feasible

actions. There is a large body of literature on efficient algorithms for linear contextual bandits with no corruptions (Abe et al., 2003; Auer, 2002; Chu et al., 2011; Dani et al., 2008; Rusmevichientong and Tsitsiklis, 2010; Abbasi-Yadkori et al., 2011; Li et al., 2019b), to mention a few. The significance of this setting lies in the fact that linear regression approaches are widely used in recommendation systems and advertising (Li et al., 2010; Jhalani et al., 2016; Deshpande and Montanari, 2012). Linear contextual bandits with adversarial corruptions is an arguably more challenging setting since most of the previous corruption-robust algorithms are based on the idea of action elimination (Lykouris et al., 2018; Gupta et al., 2019; Bogunovic et al., 2021), which is not applicable to the contextual bandits settings where the decision set is time varying and possibly infinite at each round. In Garcelon et al. (2020), it is shown that a malicious agent can force a linear contextual bandit algorithm to take any desired action $T - o(T)$ times over $T$ rounds, while applying adversarial corruptions to rewards with a cumulative cost that only grow logarithmically. This poses a big challenge for designing corruption robust algorithms for linear contextual bandits.

In this paper, we make a first attempt to study a linear contextual bandit model where an adversary can corrupt the rewards up to a corruption level $C$, which is defined as the the sum of biggest alteration the adversary made on rewards in each round. We propose a linear contextual bandits algorithm that is robust to reward corruption, dubbed multi-level optimism-in-the-face-of-uncertainty weighted learning (Multi-level OFUL). More specifically, our algorithm consists of the following novel techniques: (1) We design a multi-level partition scheme and adopt the idea of *sub-sampling* to do the robust estimation of the model parameters; (2) We maintain a cascade of *candidate confidence sets* corresponding to different corruption level (which is unknown) and randomly select a confidence set at each round to take the action; and (3) We design confidence sets that depend on the *variances of rewards*, which lead to a potentially tighter regret bound.

Our contributions are summarized as follows:

- We propose a variance-aware algorithm which is adaptive to the amount of adversarial corruptions $C$. To the best of our knowledge, it is the first algorithm for the setting of linear contextual bandits with adversarial corruptions which does not rely on the finite number of actions and other additional assumptions.

- We prove that the regret of our algorithm is in $\widetilde{O}\left(C^2 d\sqrt{\sum_{t=1}^{T} \sigma_t^2} + C^2\sqrt{dT} + CR\sqrt{dT}\right)$, where $d$ is the dimension of context vectors, $T$ is the number of rounds, $R$ is the range of noise and $\sigma_t^2, t = 1\ldots, T$ are the variances of instantaneous reward. Our regret upper bound has a multiplicative dependence on $C^2$ which indicates that our algorithm achieves a sub-linear regret when the corruption level satisfies $C = o(T^{1/4})$.

- We also derive a gap-dependent regret bound $\widetilde{O}\left(\frac{1}{\Delta} \cdot C^2 R^2 d + \frac{1}{\Delta} \cdot d^2 C^2 \max_{t \in [T]} \sigma_t^2\right)$ for our proposed algorithm, which is instance-dependent and thus leads to a better performance on good practical instances.

**Notation.** We use lower case letters to denote scalars, and use lower and upper case bold face letters to denote vectors and matrices respectively. We denote by $[n]$ the set $\{1, \ldots, n\}$. For a vector $\mathbf{x} \in \mathbb{R}^d$ and matrix $\mathbf{\Sigma} \in \mathbb{R}^{d \times d}$, a positive semi-definite matrix, we denote by $\|\mathbf{x}\|_2$ the vector's Euclidean norm and define $\|\mathbf{x}\|_{\mathbf{\Sigma}} = \sqrt{\mathbf{x}^\top \mathbf{\Sigma} \mathbf{x}}$. For two positive sequences $\{a_n\}$ and $\{b_n\}$ with $n = 1, 2, \ldots,$ we write $a_n = O(b_n)$ if there exists an absolute constant $C > 0$ such that $a_n \leq Cb_n$ holds for all $n \geq 1$ and write $a_n = \Omega(b_n)$ if there exists an absolute constant $C > 0$ such that $a_n \geq Cb_n$ holds for all $n \geq 1$. We use $\widetilde{O}(\cdot)$ to further hide the polylogarithmic factors. We use $\mathbb{1}(\cdot)$ to denote the indicator function.

## 2    Related Work

**Bandits with Adversarial Attacks:** There is a large body of literature on the problems of multi-armed bandits with adversarial corruptions. Most research in this area aims to design algorithms that achieve desirable regret bound in both stochastic multi-armed bandits and adversarial bandits, known as "the best of both worlds" guarantees (Bubeck and Slivkins, 2012; Seldin and Slivkins, 2014; Auer and Chiang, 2016; Seldin and Lugosi, 2017; Zimmert and Seldin, 2019). These works mainly focus on achieving bounds in the worst case and the case where there is no adversary. As a result, these

algorithms are either not robust to instances that moderate amount of corruptions occur, or suffer from restrictive assumptions on adversarial corruptions. Distinctive from the above line of research, Lykouris et al. (2018) focus on a variant of classic multi-armed bandit model in which each pull of an arm generates a stochastic reward that may be contaminated by an adversary before it is revealed to the player. In their work, the corruption level $C$ is defined as $C = \sum_t \max_a |r^t(a) - r_S^t(a)|$ where $r_S^t(a)$ is the stochastic reward of arm $a$ and $r^t(a)$ is the corrupted reward of arm $a$ at round $t$. They develop algorithms adaptive to the unknown corruption level, which achieves an $O(K^{1.5}C\sqrt{T})$ regret bound. Gupta et al. (2019) proposed an improved algorithm that can achieve a regret bound with only additive dependence on $C$.

On the other hand, many research efforts have also been devoted into designing adversarial attacks that cause standard algorithms to fail (Jun et al., 2018; Liu and Shroff, 2019; Gupta et al., 2019; Garcelon et al., 2020).

**Linear Bandits with Corruptions:** Li et al. (2019a) studied stochastic linear bandits with adversarial corruptions and achieved $\widetilde{O}(\frac{1}{\Delta} \cdot d^{5/2}C + \frac{1}{\Delta^2} \cdot d^6)$ regret bound where $d$ is the dimension of the context vectors, $\Delta$ is the gap between the rewards of the best and the second best action in the decision set $\mathcal{D}$. The distinction between Li et al. (2019a) and our work is that Li et al. (2019a) considers a fixed decision set $\mathcal{D}$ throughout all $T$ rounds, while we consider contextual bandits with changing decision set observed before each round. Bogunovic et al. (2021) also studied linear bandits with adversarial corruptions and considered the setting under the assumption that context vectors undergo small random perturbations, which is previously introduced by Kannan et al. (2018). Aside from the additional assumption, another major distinction in Bogunovic et al. (2021) is that the number of actions $k$ is finite and the regret bound depends on $k$ in the contextual setting with unknown corruption level $C$. Recently, Lee et al. (2021) considered corrupted linear bandits with a finite and fixed decision set and achieve an instance-independent regret of $\widetilde{O}(d\sqrt{T} + C)$. Though both their work and the work by Li et al. (2019a) focus on corrupted linear stochastic bandits, Lee et al. (2021) have a slightly different definition of regret and adopt a strong assumption on corruptions that in each round $t$, the corruptions on rewards are linear in the actions. Neu and Olkhovskaya (2020) studied linear contextual bandits with a finite decision set (i.e., $K$ actions) and an adversary. Unlike our model, they assume that the adversary can add an arbitrary noise to the loss under a limited amount $\epsilon$ and prove an $\widetilde{O}((Kd)^{\frac{1}{3}}T^{\frac{2}{3}}) + \epsilon \cdot \sqrt{d}T$ regret bound for their proposed algorithm. Kapoor et al. (2019) considered the corrupted linear contextual bandits setting under a strong assumption on corruptions that for any prefix, at most an $\eta$ fraction of the rounds are corrupted.

# 3 Preliminaries

In this paper, we study linear contextual bandits with adversarial corruptions. We will introduce our model and some basic concepts in this section.

**Corrupted linear contextual bandits.** We consider the the linear contextual bandits model studied in Abbasi-Yadkori et al. (2011) under the same corruption studied by Lykouris et al. (2018). In detail, distinctive from the linear contextual bandits Abbasi-Yadkori et al. (2011), the interaction between the agent and the environment is now contaminated by an adversary. The protocol between the agent and the adversary at each round $t \in [T]$ can be described as follows:

1. At the beginning of round $t$, the environment generates an arbitrary decision set $\mathcal{D}_t \subseteq \mathbb{R}^d$ where each element represents a feasible action that can be selected by the agent.

2. The environment generates stochastic reward function $r_t'(\mathbf{a}) = \langle \mathbf{a}, \boldsymbol{\mu}^* \rangle + \epsilon_t(\mathbf{a})$ together with an upper bound on the standard variance of $\epsilon_t(\mathbf{a})$, i.e., $\sigma_t(\mathbf{a})$ for all $\mathbf{a} \in \mathcal{D}_t$.

3. The adversary observes $\mathcal{D}_t, r_t'(\mathbf{a}), \sigma_t(\mathbf{a})$ for all $\mathbf{a} \in \mathcal{D}_t$ and decides a corrupted reward function $r_t$ defined over $\mathcal{D}_t$.

4. The agent observes $\mathcal{D}_t$ and selects $\mathbf{a}_t \in \mathcal{D}_t$.

5. The adversary observes $\mathbf{a}_t$ and then returns $r_t(\mathbf{a}_t)$ and $\sigma_t(\mathbf{a}_t)$.

6. The agent observes $r_t(\mathbf{a}_t), \sigma_t(\mathbf{a}_t)$.

Let $\mathcal{F}_t$ be the $\sigma$-algebra generated by $\mathcal{D}_{1:t}, \mathbf{a}_{1:t-1}, \epsilon_{1:t-1}, r_{1:t-1}$ and $\sigma_{1:t-1}$.

At step 2, $\boldsymbol{\mu}^*$ is a hidden vector unknown to the agent which can be observed by the adversary at the beginning. We assume that for all $t \geq 1$ and all $\mathbf{a} \in \mathcal{D}_t$, $\|\mathbf{a}\|_2 \leq A$, $|\langle \mathbf{a}, \boldsymbol{\mu}^* \rangle| \leq 1$ and $\|\boldsymbol{\mu}^*\|_2 \leq B$ almost surely. $\epsilon_t(\mathbf{a})$ can be any form of random noise as long as it satisfies

$$\forall t \geq 1, \forall \mathbf{a} \in \mathcal{D}_t, |\epsilon_t(\mathbf{a})| \leq R, \quad \mathbb{E}[\epsilon_t(\mathbf{a})|\mathcal{F}_t] = 0, \quad \mathbb{E}[\epsilon_t^2(\mathbf{a})|\mathcal{F}_t] \leq \sigma_t^2(\mathbf{a}). \tag{3.1}$$

This assumption on $\epsilon_t$ is a variant of that in Zhou et al. (2020): We now require the noise to be generated for all $\mathbf{a} \in \mathcal{D}_t$ in advance before the adversary decides the corrupted reward function. Our assumption on noises is more general than those in (Li et al., 2019a; Bogunovic et al., 2021; Kapoor et al., 2019) where they are assumed to be 1-sub-Gaussian or Gaussian.

At step 3, we assume that the adversary has observed all the previous information and thus may predict which policy the agent will take at the current round. However, since the agent can take a randomized policy, the adversary may not know exactly which action the agent will take.

**Corruption level.** We define corruption level

$$C = \frac{1}{R+1} \sum_{t=1}^{T} \sup_{\mathbf{a} \in \mathcal{D}_t} |r_t'(\mathbf{a}) - r_t(\mathbf{a})|. \tag{3.2}$$

to indicate the level of adversarial contamination. We say a model is $C$-corrupted if the corruption level is no larger than $C$.

Our definition of corruption level is equivalent to the counterpart in Lykouris et al. (2018) and Gupta et al. (2019) where they define $C = \sum_{t=1}^{T} \max_{\mathbf{a}} |r_t'(\mathbf{a}) - r_t(\mathbf{a})|$ in our notation of rewards. We introduce a factor of $\frac{1}{R+1}$ since the noise is of range $R$ in our model, while they assume all the rewards are in range $[0, 1]$.

**Regret.** Since the actions selected by the agent may not be deterministic, we define the regret for this model as follows:

$$\mathbf{Regret}(T) = \sum_{t=1}^{T} \langle \mathbf{a}_t^*, \boldsymbol{\mu}^* \rangle - \mathbb{E}\left[\sum_{t=1}^{T} \langle \mathbf{a}_t, \boldsymbol{\mu}^* \rangle\right]. \tag{3.3}$$

Our definition follows from the definition in Gupta et al. (2019) where the standard metric in stochastic multi-armed bandit models of pseudo-regret is adopted. But note that we need to take the expectation on $\sum_{t=1}^{T} r_t'(\mathbf{a}_t)$ (the second term in (3.3)), since a randomized policy is applied in each round.

**Gap.** Let $\Delta_t$ be the gap between the rewards of the best and the second best action in the decision set $\mathcal{D}_t$ as defined in Dani et al. (2008) which can be formally written as

$$\Delta_t = \min_{\mathbf{a} \in \mathcal{D}_t, \mathbf{a} \notin \mathcal{A}_t^*} \left( \langle \mathbf{a}_t^*, \boldsymbol{\mu}^* \rangle - \langle \mathbf{a}, \boldsymbol{\mu}^* \rangle \right). \tag{3.4}$$

where $\mathcal{A}_t^* = \operatorname{argmax}_{\mathbf{a} \in \mathcal{D}_t} \langle \mathbf{a}, \boldsymbol{\mu}^* \rangle$ and $\mathbf{a}_t^*$ is an arbitrary element in $\mathcal{A}_t^*$. Let $\Delta$ denotes the smallest gap $\min_{t \in [T]} \Delta_t$.

# 4 The Proposed Algorithm

In this section, we propose a variance-aware algorithm, Multi-level OFUL, in Algorithm 1, to tackle the corrupted linear contextual bandits problem. At the core of our algorithm is an action partition scheme to group historical selected actions and use them to select the future actions in different groups with different probabilities. Such a scheme is introduced to deal with the unknown corruption level. For simplicity, we denote $r_t(\mathbf{a}_t), \sigma_t(\mathbf{a}_t)$ in Section 3 by $r_t, \sigma_t$ in our algorithm.

**Main difficulty in our setting.** We begin with the main difficulty that prevents us from applying existing algorithms to our setting. Consider a simpler setting where the agent knows the corruption level $C$ in prior, and we have $\sigma_t = R$ for all $t$. Then we can apply OFUL (Abbasi-Yadkori et al., 2011) to solve our problem. In detail, in each round we estimate $\boldsymbol{\mu}^*$ by $\boldsymbol{\mu}_t$, which is the minimizer of the following ridge regression problem:

$$\boldsymbol{\mu}_t = \operatorname*{argmin}_{\boldsymbol{\mu} \in \mathbb{R}^d} \lambda \|\boldsymbol{\mu}\|_2^2 + \sum_{i=1}^{t-1} [\langle \boldsymbol{\mu}, \mathbf{a}_i \rangle - r_i]^2. \tag{4.1}$$

---

**Algorithm 1** Multi-level OFUL

1: Set the largest level of confidence sets: $\ell_{\max} \leftarrow \lceil \log_2 2T \rceil$.
2: For $\ell \in [\ell_{\max}]$, set $\boldsymbol{\Sigma}_{1,\ell} \leftarrow \lambda \mathbf{I}, \boldsymbol{\mu}_{1,\ell} \leftarrow \mathbf{0}, \mathbf{c}_{1,\ell} \leftarrow \mathbf{0}$.
3: Set $\boldsymbol{\Sigma}_1 \leftarrow \lambda \mathbf{I}, \boldsymbol{\mu}_1 \leftarrow \mathbf{0}, \mathbf{c}_1 \leftarrow \mathbf{0}$.
4: **for** $t = 1, \cdots, T$ **do**
5:     Observe $\mathcal{D}_t$.
6:     **for** $\ell = 1, \cdots, \ell_{\max}$ **do**
7:         Set $\beta_{t,\ell}$ and $\gamma_{t,\ell}$ as defined in (4.5) and (4.6).
8:         $\mathcal{C}'_{t,\ell} \leftarrow \{\boldsymbol{\mu} | \|\boldsymbol{\mu} - \boldsymbol{\mu}_t\|_{\boldsymbol{\Sigma}_t} \leq \beta_{t,\ell}\} \cap \{\boldsymbol{\mu} | \|\boldsymbol{\mu} - \boldsymbol{\mu}_{t,\ell}\|_{\boldsymbol{\Sigma}_{t,\ell}} \leq \gamma_{t,\ell}\}$.
9:         $\mathcal{C}_{t,\ell} \leftarrow \begin{cases} \mathcal{C}'_{t,\ell}, & \mathcal{C}'_{t,\ell} \neq \varnothing \\ \mathcal{C}_{t,\ell+1}, & \text{otherwise} \end{cases}$.
10:     **end for**
11:     Set $f(t) = \begin{cases} \ell & \text{with probability } 2^{-\ell} \quad 1 < \ell \leq \ell_{\max} \\ 1 & \text{otherwise} \end{cases}$.
12:     Select $\mathbf{a}_t \leftarrow \operatorname{argmax}_{\mathbf{a} \in \mathcal{D}_t} \max_{\boldsymbol{\mu} \in \mathcal{C}_{t,f(t)}} \langle \boldsymbol{\mu}, \mathbf{a} \rangle$ and observe $r_t, \sigma_t$.
13:     Set $\overline{\sigma}_t = \max\{(R+1)/\sqrt{d}, \sigma_t\}$.
14:     $\boldsymbol{\Sigma}_{t+1} \leftarrow \boldsymbol{\Sigma}_t + \mathbf{a}_t \mathbf{a}_t^\top / \overline{\sigma}_t^2, \mathbf{c}_{t+1} \leftarrow \mathbf{c}_t + r_t \mathbf{a}_t / \overline{\sigma}_t^2, \boldsymbol{\mu}_{t+1} \leftarrow \boldsymbol{\Sigma}_{t+1}^{-1} \mathbf{c}_{t+1}$.
15:     **for** $\ell \neq f(t)$ **do**
16:         $\boldsymbol{\Sigma}_{t+1,\ell} \leftarrow \boldsymbol{\Sigma}_{t,\ell}, \mathbf{c}_{t+1,\ell} \leftarrow \mathbf{c}_{t,\ell}, \boldsymbol{\mu}_{t+1,\ell} \leftarrow \boldsymbol{\mu}_{t,\ell}$.
17:     **end for**
18:     $\boldsymbol{\Sigma}_{t+1,f(t)} \leftarrow \boldsymbol{\Sigma}_{t,f(t)} + \mathbf{a}_t \mathbf{a}_t^\top / \overline{\sigma}_t^2, \mathbf{c}_{t+1,f(t)} \leftarrow \mathbf{c}_{t,f(t)} + r_t \mathbf{a}_t / \overline{\sigma}_t^2$.
19:     $\boldsymbol{\mu}_{t+1,f(t)} \leftarrow \boldsymbol{\Sigma}_{t+1,f(t)}^{-1} \mathbf{c}_{t+1,f(t)}$.
20: **end for**

---

By slightly modifying the self-normalized martingale concentration inequality proposed in Abbasi-Yadkori et al. (2011), we can conclude that $\boldsymbol{\mu}^*$ belongs to the ellipsoid $\|\boldsymbol{\mu} - \boldsymbol{\mu}_t\|_{\boldsymbol{\Sigma}_t^{-1}} \leq \beta_t$ with high probability, where $\beta_t = \widetilde{O}(R\sqrt{d} + C\sqrt{d})$. Such a confidence bound leads to a final regret which has a polynomial dependence on $C$. However, such a simple approach have two limitations. First, the agent does not know $C$ apriori in our setting, thus it is impossible to set $\beta_t$ to be dependent on $C$. Second, vanilla ridge regression estimator does not consider different variances $\sigma_t$ in each round, thus it only gives a very conservative estimation.

**Action partition scheme.** To address the unknown $C$ issue, besides the original estimator $\boldsymbol{\mu}_t$ which uses all previous data, Algorithm 1 maintains several additional learners to learn $\boldsymbol{\mu}^*$ at different accuracy level simultaneously, and it *randomly* selects one of the learners with different probabilities at each round. Such a "parallel learning" idea is inspired by Lykouris et al. (2018). In detail, we partition the observed data into $\ell_{\max}$ levels indexed by $[\ell_{\max}]$ and maintain $\ell_{\max}$ sub-sampled estimators $\boldsymbol{\mu}_{t,1}, \cdots, \boldsymbol{\mu}_{t,\ell_{\max}}$. According to line 11, the observed data in round $t$ goes into level $\ell$ with probability $2^{-\ell}$ if $1 < \ell \leq \ell_{\max}$ and it goes to level 1 with probability $1 - \sum_{\ell=2}^{\ell_{\max}} 2^{-\ell} = 1/2 + 2^{-\ell_{\max}}$. The intuition is that if $2^\ell \geq C$, then the corruption level experienced by level $\ell$

$$\text{Corruption}_{t,\ell} = \sum_{i=1}^{t} \frac{\mathbb{1}(f(i) = \ell)}{R+1} \cdot \sup_{\mathbf{a} \in \mathcal{D}_i} |r_i(\mathbf{a}) - r'_i(\mathbf{a})| \tag{4.2}$$

can be bounded by some quantity that is *independent of $C$*. That says, the individual learners whose level is greater than $\log C$ can learn $\boldsymbol{\mu}^*$ successfully, even with the corruption. For the learners whose level is less than $\log C$, we can also control the error by controlling the probability for the agent to select them.

**Weighted regression estimator.** After introducing the partition scheme, we still need to deal with the varying variance (heteroscedastic) case. Similar to (Kirschner and Krause, 2018; Zhou et al., 2020), we proposed the following *weighted ridge regression estimator*, which incorporates the variance information of the rewards into estimation:

$$\boldsymbol{\mu}_t = \operatorname*{argmin}_{\boldsymbol{\mu} \in \mathbb{R}^d} \lambda \|\boldsymbol{\mu}\|_2^2 + \sum_{i=1}^{t-1} [\langle \boldsymbol{\mu}, \mathbf{a}_i \rangle - r_i]^2 / \overline{\sigma}_i^2. \tag{4.3}$$

Here $\overline{\sigma}_t$ is defined as the upper bound of the true variance $\sigma_t$ in line 13. The closed-form solution to (4.3) is calculated at each round in line 14. The use of $\overline{\sigma}_t$, as we will show later, makes our estimator more efficient in the heteroscedastic case. Meanwhile, we also apply our weighted regression estimator to each individual learner, and their estimator $\boldsymbol{\mu}_{t,\ell}$ can be written as follows:

$$\boldsymbol{\mu}_{t,\ell} = \underset{\boldsymbol{\mu} \in \mathbb{R}^d}{\operatorname{argmin}} \, \lambda \|\boldsymbol{\mu}\|_2^2 + \sum_{i=1}^{t-1} \mathbb{1}(f(i) = \ell) \cdot [\langle \boldsymbol{\mu}, \mathbf{a}_i \rangle - r_i]^2 / \overline{\sigma}_i^2. \quad (4.4)$$

The closed-form solution to (4.4) is calculated at each round in lines 15–20.

**Final Multi-Level confidence sets.** With the estimators $\boldsymbol{\mu}_t, \boldsymbol{\mu}_{t,1}, \cdots, \boldsymbol{\mu}_{t,\ell_{\max}}$ at the beginning of round $t$, we define a cascade of candidate confidence sets as in lines 6–10, where

$$\beta_{t,\ell} = 8\sqrt{d \log \frac{(R+1)^2\lambda + tA^2}{(R+1)^2\lambda} \log(4t^2/\delta)} + 4\sqrt{d}\log(4t^2/\delta) + 2^\ell\sqrt{d} + \sqrt{\lambda}B, \quad (4.5)$$

$$\gamma_{t,\ell} = 8\sqrt{d \log \frac{(R+1)^2\lambda + tA^2}{(R+1)^2\lambda} \log(8t^2T/\delta)} + 4\sqrt{d}\log(8t^2T/\delta) + \overline{C}_\ell\sqrt{d} + \sqrt{\lambda}B, \quad (4.6)$$

with $\overline{C}_\ell = \log(2\ell^2/\delta) + 3$. For simplicity, we define

$$\ell^* = \max\{2, \lceil \log_2 C \rceil\} \quad (4.7)$$

as an important threshold in our later proof for regret bound analysis. Later we will prove that $\mathcal{C}_{t,\ell}$ contains $\boldsymbol{\mu}^*$ for all $\ell \geq \ell^*$, $t \geq 1$ with high probability.

Note that each candidate confidence set can be written as the intersection of two ellipsoids. The intuition behind our construction of candidate confidence sets is that we hope that $\mathcal{C}_{t,\ell}$ is robust enough to handle the $2^\ell$-corrupted case, i.e., $\boldsymbol{\mu}^* \in \mathcal{C}_{t,\ell}$ with high probability. To achieve this, the first ellipsoid makes use of the global information and the "radius" $\beta_{t,\ell}$ need to contain a factor of $2^\ell$ to tolerate a corruption level of $2^\ell$, and the second ellipsoid makes use of the observed data in level $\ell$ since this level only contain a few times of corruptions in $2^\ell$-corrupted case.

**Action selection.** With the candidate confidence sets, we use line 11 to randomly decide one confidence set and select an action based on the optimism-in-the-face-of-uncertainty (OFU) principle in line 12. Then we update the estimators for the next round $t + 1$.

**Remark 4.1.** Our algorithm shares a similar strategy for partitioning the observed data with the algorithm in Lykouris et al. (2018) but note that there is a major difference in that: Lykouris et al. (2018) regard the partition scheme as a "layer structure", i.e., their algorithm further uses different estimators in layers of parallel learners and do action elimination layer by layer in each round. In contrast, the sub-sampled estimators in our algorithm are used independently, i.e., the selected action only relies on one of the partitions. As a result, Algorithm 1 does not need to do action elimination, thus is capable of handling the cases where the number of actions is huge or even infinite.

## 5 Main Results

In this section we present our main theorem, which establishes the regret bound for Multi-level OFUL.

**Theorem 5.1.** Set $\lambda = 1/B^2$. Suppose that $C = \Omega(1)$, $R = \Omega(1)$, for all $t \geq 1$ and all $\mathbf{a} \in \mathcal{D}_t$, $\langle \mathbf{a}, \boldsymbol{\mu}^* \rangle \in [-1, 1]$. Then with probability at least $1 - 3\delta$, the regret of Algorithm 1 is bounded as follows:

$$\mathbf{Regret}(T) = \widetilde{O}\left(C^2 d \sqrt{\sum_{t=1}^{T} \sigma_t^2} + C^2\sqrt{dT} + CR\sqrt{dT}\right).$$

**Remark 5.2.** When $\sigma_t, R = \Omega(1)$, the regret bound in Theorem 5.1 matches the regret bound of OFUL proposed in Zhou et al. (2020) when the corruption level $C$ is a constant.

**Remark 5.3.** Compared with the $\widetilde{O}(d\sqrt{T} + C)$ result in Lee et al. (2021), our result has a multiplicative quadratic dependence on $C$, which seems to be worse. However, we want to emphasize that we focus on a more challenging contextual bandits setting where the decision sets $\mathcal{D}_t$ at each round are not identical, which is different from that in Lee et al. (2021). Therefore, our result and that in Lee et al. (2021) are not directly comparable.

**Remark 5.4.** Note that this instance-independent regret upper bound also holds in a stronger model than the one described in Section 3, where the adversary can even decide the decision set $\mathcal{D}_t$ at each round $t$ since our regret bound can hold without any assumption on the decision sets.

**Corollary 5.5.** Under the same conditions as in Theorem 5.1, if $\sigma_t$ given by the environment are all $R$, the regret of Algorithm 1 is bounded by:

$$\mathbf{Regret}(T) = \widetilde{O}\left(C^2 dR\sqrt{T}\right).$$

We also provide a gap-dependent regret bound.

**Theorem 5.6.** Suppose that $C = \Omega(1)$, $R = \Omega(1)$, for all $t \geq 1$ and all $\mathbf{a} \in \mathcal{D}_t$, $\langle \mathbf{a}, \boldsymbol{\mu}^* \rangle \in [-1, 1]$. Then with probability at least $1 - 3\delta$, the regret of Algorithm 1 is bounded as follows:

$$\mathbf{Regret}(T) = \widetilde{O}\left(\frac{1}{\Delta} \cdot C^2 R^2 d + \frac{1}{\Delta} \cdot d^2 C^2 \max_{t \in [T]} \sigma_t^2\right).$$

**Remark 5.7.** Theorem 5.6 automatically suggests an $\widetilde{O}(R^2 d^2 C^2/\Delta)$ regret bound, by the fact $\sigma_t = O(R)$. Compared with previous result $\widetilde{O}(d^{5/2}C/\Delta + d^6/\Delta^2)$ (Lee et al., 2021), our result has a better dependence on the dimension $d$ but a worse dependence on the corruption level $C$. As Remark 5.3 suggests, we focus on a more challenging contextual bandits setting, and the worse dependence on $C$ might be due to this.

# 6 Proof Outline

First we have the following lemma which is a corruption-tolerant variant of Bernstein inequality for self-normalized vector-valued martingales introduced in Zhou et al. (2020).

**Lemma 6.1** (Bernstein inequality for vector-valued martingales with corruptions). Let $\{\mathcal{G}_t\}_{t=1}^{\infty}$ be a filtration, $\{\mathbf{x}_t, \eta_t\}_{t\geq 1}$ a stochastic process so that $\mathbf{x}_t \in \mathbb{R}^d$ is $\mathcal{G}_t$-measurable and $\eta_t \in \mathbb{R}$ is $\mathcal{G}_{t+1}$-measurable. Fix $R, L, \sigma, \lambda > 0$, $\boldsymbol{\mu}^* \in \mathbb{R}^d$. For $t \geq 1$ let $y_t^{\text{stoch}} = \langle \boldsymbol{\mu}^*, \mathbf{x}_t \rangle + \eta_t$ and suppose that $\eta_t, \mathbf{x}_t$ also satisfy

$$|\eta_t| \leq R, \mathbb{E}[\eta_t | \mathcal{G}_t] = 0, \mathbb{E}[\eta_t^2 | \mathcal{G}_t] \leq \sigma^2, \|\mathbf{x}_t\|_2 \leq L.$$

Suppose $\{y_t\}$ is a sequence such that $\sum_{i=1}^t |y_i - y_i^{\text{stoch}}| = C(t)$ for all $t \geq 1$. Then, for any $0 < \delta < 1$, with probability at least $1 - \delta$ we have $\forall t > 0$,

$$\|\boldsymbol{\mu}_t - \boldsymbol{\mu}^*\|_{\mathbf{Z}_t} \leq \beta_t + C(t) + \sqrt{\lambda}\|\boldsymbol{\mu}^*\|_2,$$

where for $t \geq 1$, $\boldsymbol{\mu}_t = \mathbf{Z}_t^{-1}\mathbf{b}_t$, $\mathbf{Z}_t = \lambda\mathbf{I} + \sum_{i=1}^t \mathbf{x}_i\mathbf{x}_i^\top$, $\mathbf{b}_t = \sum_{i=1}^t y_i\mathbf{x}_i$, and

$$\beta_t = 8\sigma\sqrt{d\log\frac{d\lambda + tL^2}{d\lambda}\log(4t^2/\delta)} + 4R\log(4t^2/\delta).$$

Next, we have that with high probability, all the level $\ell \geq \ell^*$ only influenced by limited amount of corruptions as mentioned in Section 4.

**Lemma 6.2.** Let $\text{Corruption}_{t,\ell}$ be defined in (4.2). Then we have with probability at least $1 - \delta$, for all $\ell \geq \ell^*$, $t \geq 1$:

$$\text{Corruption}_{t,\ell} \leq \overline{C}_\ell = \log(2\ell^2/\delta) + 3.$$

We denote by $\mathcal{E}_{\text{sub}}$ the event that the above inequality holds.

We define the following event to further show that our candidate confidence sets with $\ell \geq \ell^*$ are "robust" enough, i.e. $\mathcal{C}_{t,\ell}$ contains $\boldsymbol{\mu}^*$ with high probability.

**Definition 6.3.** Let $\ell^*$ be defined in (4.7). We introduce the event $\mathcal{E}_1$ as follows.

$$\mathcal{E}_1 := \left\{\forall\ell \geq \ell^* \text{ and } t \geq 1, \|\boldsymbol{\mu}^* - \boldsymbol{\mu}_t\|_{\boldsymbol{\Sigma}_t} \leq \beta_{t,\ell} \text{ and } \|\boldsymbol{\mu}^* - \boldsymbol{\mu}_{t,\ell}\|_{\boldsymbol{\Sigma}_{t,\ell}} \leq \gamma_{t,\ell}\right\}. \quad (6.1)$$

where $\beta_{t,\ell}, \gamma_{t,\ell}$ are defined in (4.5) and (4.6).

254 Next lemma suggests that the event $\mathcal{E}_1$ happens with high probability.

255 **Lemma 6.4.** Let $\mathcal{E}_1$ be defined in (6.1). For any $0 < \delta < 1/3$, we have $\mathbb{P}(\mathcal{E}_1) \geq 1 - 3\delta$.

256 For simplicity, we define $\mathbf{a}_{t,\ell} = \mathrm{argmax}_{\mathbf{a} \in \mathcal{D}_t} \max_{\boldsymbol{\mu} \in \mathcal{C}_{t,\ell}} \langle \boldsymbol{\mu}, \mathbf{a} \rangle$ for each level $\ell$. $\mathbf{a}_t$ can be seen as an
257 action vector randomly chosen from $\mathbf{a}_{t,\ell}$, $\ell \in [\ell_{\max}]$. Next two lemmas suggest that under event $\mathcal{E}_1$,
258 at each round, the gap between the optimal reward and the selected reward can be upper bounded by
259 some bonus terms related to $\mathbf{a}_{t,\ell}$.

260 **Lemma 6.5.** Suppose $\mathcal{E}_1$ occurs. If $f(t) \leq \ell^*$, we have $\langle \mathbf{a}_t^* - \mathbf{a}_t, \boldsymbol{\mu}^* \rangle \leq 2\beta_{t,\ell^*} \|\mathbf{a}_t\|_{\boldsymbol{\Sigma}_t^{-1}} +$
261 $2\beta_{t,\ell^*} \|\mathbf{a}_{t,\ell^*}\|_{\boldsymbol{\Sigma}_t^{-1}}$.

262 **Lemma 6.6.** On event $\mathcal{E}_1$, if $f(t) = \ell > \ell^*$, we have $\langle \mathbf{a}_t^* - \mathbf{a}_t, \boldsymbol{\mu}^* \rangle \leq 2\gamma_{t,\ell} \|\mathbf{a}_t\|_{\boldsymbol{\Sigma}_{t,\ell}^{-1}}$.

263 Now we provide the proof sketch of Theorem 5.1.

264 *Proof sketch of Theorem 5.1 .* Suppose $\mathcal{E}_1$ occurs. The main idea to bound the regret is to decompose
265 the total rounds $[T]$ into two non-overlapping parts, based on which individual learner is selected at
266 that round. In detail, we have

$$
\begin{aligned}
\mathbf{Regret}(T) &= \mathbb{E}\left[\sum_{t=1}^{T} (\langle \mathbf{a}_t^*, \boldsymbol{\mu}^* \rangle - \langle \mathbf{a}_t, \boldsymbol{\mu}^* \rangle)\right] \\
&= \mathbb{E}\underbrace{\left[\sum_{t=1}^{T} \mathbb{1}(f(t) \leq \ell^*) (\langle \mathbf{a}_t^*, \boldsymbol{\mu}^* \rangle - \langle \mathbf{a}_t, \boldsymbol{\mu}^* \rangle)\right]}_{I_1} \\
&\quad + \sum_{\ell=\ell^*+1}^{\ell_{\max}} \mathbb{E}\underbrace{\left[\sum_{t=1}^{T} \mathbb{1}(f(t) = \ell) (\langle \mathbf{a}_t^*, \boldsymbol{\mu}^* \rangle - \langle \mathbf{a}_t, \boldsymbol{\mu}^* \rangle)\right]}_{I_2(\ell)}.
\end{aligned}
\tag{6.2}
$$

267 Here $I_1$ represents the regret where the the "low-level" learner is selected, where the corruption level
268 is beyond the learner level. For this case, by Lemma 6.5, we can directly show that

$$
I_1 \leq \mathbb{E}\left[\sum_{t=1}^{T} \mathbb{1}(f(t) \leq \ell^*) \min\left\{2, 2\beta_{t,\ell^*} \|\mathbf{a}_{t,\ell^*}\|_{\boldsymbol{\Sigma}_t^{-1}} + 2\beta_{t,\ell^*} \|\mathbf{a}_t\|_{\boldsymbol{\Sigma}_t^{-1}}\right\}\right].
\tag{6.3}
$$

269 We further bound (6.3). Let $\mathcal{F}_t$ be the $\sigma$-algebra generated by $\mathbf{a}_s, r_s, \sigma_s, f(s)$ for $s \leq t-1$.
270 Then by the property of our partition scheme (note that $\mathbb{P}(f(t) = \ell^*) = 2^{-\ell^*}$), we can show that
271 $\mathbb{E}\left[\mathbb{1}(f(t) \leq \ell^*) \|\mathbf{a}_{t,\ell^*}\|_{\boldsymbol{\Sigma}_t^{-1}} | \mathcal{F}_t\right] \leq 2^{\ell^*} \mathbb{E}\left[\|\mathbf{a}_t\|_{\boldsymbol{\Sigma}_t^{-1}} | \mathcal{F}_t\right]$. Therefore, we can further bound $I_1$ by

$$
I_1 \leq 4 \cdot 2^{\ell^*} \mathbb{E}\underbrace{\left[\sum_{t=1}^{T} \min\left\{2, \beta_{T,\ell^*} \|\mathbf{a}_t\|_{\boldsymbol{\Sigma}_t^{-1}}\right\}\right]}_{I_3}.
\tag{6.4}
$$

272 To further bound $I_3$, we split $[T]$ into 2 parts, $\mathcal{I}_1 = \{t \in [T] | \|\mathbf{a}_t/\overline{\sigma}_t\|_{\boldsymbol{\Sigma}_t^{-1}} > 1\}, \mathcal{I}_2 = \{t \in
273 [T] | \|\mathbf{a}_t/\overline{\sigma}_t\|_{\boldsymbol{\Sigma}_t^{-1}} \leq 1\}$ to bound $I_3$. The intuition here is that the cardinality of $\mathcal{I}_1$ is bounded, and
274 the sum of terms with $t \in \mathcal{I}_2$ can be bounded using Cauchy-Schwarz inequality.

$$
\sum_{t \in \mathcal{I}_1} \min\left\{2, \beta_{T,\ell^*} \|\mathbf{a}_t\|_{\boldsymbol{\Sigma}_t^{-1}}\right\} \leq 2|\mathcal{I}_1| \leq 2\sum_{t=1}^{T} \min\left\{1, \|\mathbf{a}_t/\overline{\sigma}_t\|_{\boldsymbol{\Sigma}_t^{-1}}^2\right\} \leq 4d \log \frac{(R+1)^2\lambda + TA^2}{(R+1)^2\lambda},
\tag{6.5}
$$

275 where the first inequality holds since $\min\left\{2, \beta_{T,\ell^*} \|\mathbf{a}_t\|_{\boldsymbol{\Sigma}_t^{-1}}\right\} \leq 2$, the second inequality follows
276 from the definition of $\mathcal{I}_1$, the third inequality holds by Lemma C.2.

$$\sum_{t \in \mathcal{I}_2} \min \left\{ 2, \beta_{T,\ell^*} \|\mathbf{a}_t\|_{\boldsymbol{\Sigma}_t^{-1}} \right\} \leq \beta_{T,\ell^*} \sqrt{\sum_{t \in \mathcal{I}_2} \overline{\sigma}_t^2} \cdot \sqrt{\sum_{t \in \mathcal{I}_2} \min \left\{ 1, \|\mathbf{a}_t/\overline{\sigma}_t\|_{\boldsymbol{\Sigma}_t^{-1}}^2 \right\}}$$

$$\leq \beta_{T,\ell^*} \sqrt{(R+1)^2 T/d + \sum_{t=1}^{T} \sigma_t^2} \cdot \sqrt{2d \log \frac{(R+1)^2 \lambda + T A^2}{(R+1)^2 \lambda}}, \tag{6.6}$$

where the first inequality follows from Cauchy-Schwarz inequality, the second inequality follows from the definition of $\overline{\sigma}_t$ and Lemma C.2.

Substituting (6.5) and (6.6) into (6.3), we have

$$I_1 = \widetilde{O} \left( C^2 d \sqrt{\sum_{t=1}^{T} \sigma_t^2} + C^2 \sqrt{dT} + CR\sqrt{dT} \right). \tag{6.7}$$

Now it remains to bound $I_2(\ell)$. By Lemma 6.6, we have

$$I_2(\ell) \leq 2\mathbb{E} \underbrace{\left[ \sum_{t=1}^{T} \mathbb{1}(f(t) = \ell) \min \left\{ 1, \gamma_{t,\ell} \|\mathbf{a}_{t,\ell}\|_{\boldsymbol{\Sigma}_{t,\ell}^{-1}} \right\} \right]}_{I_4} = \widetilde{O} \left( R\sqrt{Td} + d\sqrt{\sum_{t=1}^{T} \sigma_t^2} \right), \tag{6.8}$$

where the second equality can be proved by analysis similar to that of (6.5) and (6.6). Finally, substituting (6.7) and (6.8) into (6.2) ends our proof.

$\square$

## 7 Conclusion and Future Work

In this paper, we have considered the linear contextual bandits problem in the presence of adversarial corruptions. We propose a Multi-level OFUL algorithm, which is provably robust to the adversarial attacks. We prove a gap-independent regret bound of $\widetilde{O} \left( C^2 d \sqrt{\sum_{t=1}^{T} \sigma_t^2} + C^2 \sqrt{dT} + CR\sqrt{dT} \right)$ together with a gap-dependent bound of $\widetilde{O} \left( \frac{1}{\Delta} \cdot C^2 R^2 d + \frac{1}{\Delta} \cdot d^2 C^2 \max_{t \in [T]} \sigma_t^2 \right)$.

We leave it as an open question that whether the multiplicative dependence on $C^2$ in the regret upper bounds can be removed without making additional assumptions in our setting.

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
