# A Proof of the Main Results

## A.1 Proof of Theorem 5.1

We first prove the following lemma which is a corruption-tolerant variant of Bernstein inequality for self-normalized vector-valued martingales introduced in Zhou et al. (2020).

**Lemma A.1** (Restatement of Lemma 6.1). Let $\{\mathcal{G}_t\}_{t=1}^{\infty}$ be a filtration, $\{\mathbf{x}_t, \eta_t\}_{t \geq 1}$ a stochastic process so that $\mathbf{x}_t \in \mathbb{R}^d$ is $\mathcal{G}_t$-measurable and $\eta_t \in \mathbb{R}$ is $\mathcal{G}_{t+1}$-measurable. Fix $R, L, \sigma, \lambda > 0$, $\boldsymbol{\mu}^* \in \mathbb{R}^d$. For $t \geq 1$ let $y_t^{\text{stoch}} = \langle \boldsymbol{\mu}^*, \mathbf{x}_t \rangle + \eta_t$ and suppose that $\eta_t, \mathbf{x}_t$ also satisfy

$$|\eta_t| \leq R, \mathbb{E}[\eta_t | \mathcal{G}_t] = 0, \mathbb{E}[\eta_t^2 | \mathcal{G}_t] \leq \sigma^2, \|\mathbf{x}_t\|_2 \leq L.$$

Suppose $\{y_t\}$ is a sequence such that $\sum_{i=1}^{t} |y_i - y_i^{\text{stoch}}| = C(t)$ for all $t \geq 1$. Then, for any $0 < \delta < 1$, with probability at least $1 - \delta$ we have $\forall t > 0$,

$$\|\boldsymbol{\mu}_t - \boldsymbol{\mu}^*\|_{\mathbf{Z}_t} \leq \beta_t + C(t) + \sqrt{\lambda} \|\boldsymbol{\mu}^*\|_2,$$

where for $t \geq 1$, $\boldsymbol{\mu}_t = \mathbf{Z}_t^{-1} \mathbf{b}_t$, $\mathbf{Z}_t = \lambda \mathbf{I} + \sum_{i=1}^{t} \mathbf{x}_i \mathbf{x}_i^{\top}$, $\mathbf{b}_t = \sum_{i=1}^{t} y_i \mathbf{x}_i$, and

$$\beta_t = 8\sigma \sqrt{d \log \frac{d\lambda + tL^2}{d\lambda} \log(4t^2/\delta)} + 4R \log(4t^2/\delta).$$

*Proof.* See Appendix B.1. $\qquad\square$

Then we prove that with high probability, all the level $\ell \geq \ell^*$ only influenced by limited amount of corruptions as mentioned in Section 4.

**Lemma A.2** (Restatement of Lemma 6.2). Let $\text{Corruption}_{t,\ell}$ be defined in (4.2). Then we have with probability at least $1 - \delta$, for all $\ell \geq \ell^*, t \geq 1$:

$$\text{Corruption}_{t,\ell} \leq \overline{C}_\ell = \log(2\ell^2/\delta) + 3.$$

We denote by $\mathcal{E}_{\text{sub}}$ the event that the above inequality holds.

*Proof.* The proof of this lemma is based on Lemma B.1 introduced by Lykouris et al. (2018); for details see Appendix B.2. $\qquad\square$

We define the following event to further show that our candidate confidence sets with $\ell \geq \ell^*$ are "robust" enough, i.e. $\mathcal{C}_{t,\ell}$ contains $\boldsymbol{\mu}^*$ with high probability.

**Definition A.3.** Let $\ell^*$ be defined in (4.7). We introduce the event $\mathcal{E}_1$ as follows.

$$\mathcal{E}_1 := \left\{ \forall \ell \geq \ell^* \text{ and } t \geq 1, \|\boldsymbol{\mu}^* - \boldsymbol{\mu}_t\|_{\boldsymbol{\Sigma}_t} \leq \beta_{t,\ell} \text{ and } \|\boldsymbol{\mu}^* - \boldsymbol{\mu}_{t,\ell}\|_{\boldsymbol{\Sigma}_{t,\ell}} \leq \gamma_{t,\ell} \right\}. \tag{A.1}$$

Recall that

$$\beta_{t,\ell} = 8\sqrt{d \log \frac{(R+1)^2 \lambda + tA^2}{(R+1)^2 \lambda} \log(4t^2/\delta)} + 4\sqrt{d} \log(4t^2/\delta) + 2^\ell \sqrt{d} + \sqrt{\lambda} B, \tag{A.2}$$

$$\gamma_{t,\ell} = 8\sqrt{d \log \frac{(R+1)^2 \lambda + tA^2}{(R+1)^2 \lambda} \log(8t^2 T/\delta)} + 4\sqrt{d} \log(8t^2 T/\delta) + \overline{C}_\ell \sqrt{d} + \sqrt{\lambda} B. \tag{A.3}$$

**Lemma A.4** (Restatement of Lemma 6.4). Let $\mathcal{E}_1$ be defined in (A.1). For any $0 < \delta < 1$, we have $\mathbb{P}(\mathcal{E}_1) \geq 1 - 3\delta$.

*Proof.* See Appendix B.3. $\qquad\square$

**Definition A.5.** For simplicity, we define $\mathbf{a}_{t,\ell} = \text{argmax}_{\mathbf{a} \in \mathcal{D}_t} \max_{\boldsymbol{\mu} \in \mathcal{C}_{t,\ell}} \langle \boldsymbol{\mu}, \mathbf{a} \rangle$ for each level $\ell$.

With this definition, $\mathbf{a}_t$ can be seen as an action vector randomly chosen from $\mathbf{a}_{t,\ell}$, $\ell \in [\ell_{\max}]$. In the following part of this section, we show how to derive the instance-independent regret upper bound using this notation.

**Lemma A.6** (Restatement of Lemms 6.5). Suppose $\mathcal{E}_1$ occurs. If $f(t) \leq \ell^*$, we have $\langle \mathbf{a}_t^* - \mathbf{a}_t, \boldsymbol{\mu}^* \rangle \leq 2\beta_{t,\ell^*}\|\mathbf{a}_t\|_{\boldsymbol{\Sigma}_t^{-1}} + 2\beta_{t,\ell^*}\|\mathbf{a}_{t,\ell^*}\|_{\boldsymbol{\Sigma}_t^{-1}}$.

**Lemma A.7** (Restatement of Lemms 6.6). On event $\mathcal{E}_1$, if $f(t) = \ell > \ell^*$, we have $\langle \mathbf{a}_t^* - \mathbf{a}_t, \boldsymbol{\mu}^* \rangle \leq 2\gamma_{t,\ell}\|\mathbf{a}_t\|_{\boldsymbol{\Sigma}_{t,\ell}^{-1}}$.

*Proof of Theorem 5.1.* Suppose $\mathcal{E}_1$ occurs. We divide regret into two parts,

$$
\begin{aligned}
\mathbf{Regret}(T) &= \mathbb{E}\left[\sum_{t=1}^{T} (\langle \mathbf{a}_t^*, \boldsymbol{\mu}^* \rangle - \langle \mathbf{a}_t, \boldsymbol{\mu}^* \rangle)\right] \\
&= \mathbb{E}\underbrace{\left[\sum_{t=1}^{T} \mathbb{1}(f(t) \leq \ell^*)(\langle \mathbf{a}_t^*, \boldsymbol{\mu}^* \rangle - \langle \mathbf{a}_t, \boldsymbol{\mu}^* \rangle)\right]}_{I_1} \\
&\quad + \sum_{\ell=\ell^*+1}^{\ell_{\max}} \mathbb{E}\underbrace{\left[\sum_{t=1}^{T} \mathbb{1}(f(t) = \ell)(\langle \mathbf{a}_t^*, \boldsymbol{\mu}^* \rangle - \langle \mathbf{a}_t, \boldsymbol{\mu}^* \rangle)\right]}_{I_2(\ell)},
\end{aligned}
\tag{A.4}
$$

where the first equality holds by definition in (3.3).

By Lemma B.3, we have

$$
I_1 \leq \mathbb{E}\left[\sum_{t=1}^{T} \mathbb{1}(f(t) \leq \ell^*)\min\left\{2, 2\beta_{t,\ell^*}\|\mathbf{a}_{t,\ell^*}\|_{\boldsymbol{\Sigma}_t^{-1}} + 2\beta_{t,\ell^*}\|\mathbf{a}_t\|_{\boldsymbol{\Sigma}_t^{-1}}\right\}\right].
\tag{A.5}
$$

Let $\mathcal{F}_t$ be the $\sigma$-algebra generated by $\mathbf{a}_s, r_s, \sigma_s, f(s)$ for $s \leq t-1$. Note that

$$
\begin{aligned}
\mathbb{E}\left[\mathbb{1}(f(t) \leq \ell^*)\|\mathbf{a}_{t,\ell^*}\|_{\boldsymbol{\Sigma}_t^{-1}}\Big|\mathcal{F}_t\right] &= \mathbb{P}(f(t) \leq \ell^*)\|\mathbf{a}_{t,\ell^*}\|_{\boldsymbol{\Sigma}_t^{-1}} \\
&\leq 2^{\ell^*}\mathbb{P}(f(t) = \ell^*)\|\mathbf{a}_{t,\ell^*}\|_{\boldsymbol{\Sigma}_t^{-1}} \\
&\leq 2^{\ell^*}\mathbb{E}\left[\|\mathbf{a}_t\|_{\boldsymbol{\Sigma}_t^{-1}}\Big|\mathcal{F}_t\right],
\end{aligned}
\tag{A.6}
$$

where the first equality holds since $\mathbf{a}_{t,\ell^*}$ and $\boldsymbol{\Sigma}_t$ is deterministic given $\mathcal{F}_t$, the first inequality holds since $\mathbb{P}(f(t) = \ell^*) = 2^{-\ell^*}$, the last inequality holds due to the fact that $\mathbb{P}(f(t) = \ell^*)\|\mathbf{a}_{t,\ell^*}\|_{\boldsymbol{\Sigma}_t^{-1}} = \mathbb{E}\left[\mathbb{1}(f(t) = \ell^*)\|\mathbf{a}_t\|_{\boldsymbol{\Sigma}_t^{-1}}\Big|\mathcal{F}_t\right]$.

Substituting (A.6) into (A.5), we have

$$
I_1 \leq 2^{\ell^*}\mathbb{E}\left[\sum_{t=1}^{T} 4\min\{\beta_{t,\ell^*}\|\mathbf{a}_t\|_{\boldsymbol{\Sigma}_t^{-1}}, 2\}\right] \leq 4 \cdot 2^{\ell^*}\mathbb{E}\underbrace{\left[\sum_{t=1}^{T}\min\left\{2, \beta_{T,\ell^*}\|\mathbf{a}_t\|_{\boldsymbol{\Sigma}_t^{-1}}\right\}\right]}_{I_3}
\tag{A.7}
$$

We split $[T]$ into 2 parts to bound $I_3$.

Let $\mathcal{I}_1 = \{t \in [T]|\|\mathbf{a}_t/\overline{\sigma}_t\|_{\boldsymbol{\Sigma}_t^{-1}} > 1\}$, $\mathcal{I}_2 = \{t \in [T]|\|\mathbf{a}_t/\overline{\sigma}_t\|_{\boldsymbol{\Sigma}_t^{-1}} \leq 1\}$.

$$
\sum_{t \in \mathcal{I}_1}\min\left\{2, \beta_{T,\ell^*}\|\mathbf{a}_t\|_{\boldsymbol{\Sigma}_t^{-1}}\right\} \leq 2|\mathcal{I}_1| \leq 2\sum_{t=1}^{T}\min\left\{1, \|\mathbf{a}_t/\overline{\sigma}_t\|_{\boldsymbol{\Sigma}_t^{-1}}^2\right\} \leq 4d\log\frac{(R+1)^2\lambda + TA^2}{(R+1)^2\lambda},
\tag{A.8}
$$

where the first inequality holds since $\min\left\{2, \beta_{T,\ell^*}\|\mathbf{a}_t\|_{\mathbf{\Sigma}_t^{-1}}\right\} \le 2$, the second inequality follows from the definition of $\mathcal{I}_1$, the third inequality holds by Lemma C.2.

$$
\begin{aligned}
\sum_{t\in\mathcal{I}_2}\min\left\{2, \beta_{T,\ell^*}\|\mathbf{a}_t\|_{\mathbf{\Sigma}_t^{-1}}\right\} &\le \beta_{T,\ell^*}\sqrt{\sum_{t\in\mathcal{I}_2}\overline{\sigma}_t^2} \cdot \sqrt{\sum_{t\in\mathcal{I}_2}\min\left\{1, \|\mathbf{a}_t/\overline{\sigma}_t\|_{\mathbf{\Sigma}_t^{-1}}^2\right\}} \\
&\le \beta_{T,\ell^*}\sqrt{(R+1)^2 T/d + \sum_{t=1}^T \sigma_t^2} \cdot \sqrt{2d\log\frac{(R+1)^2\lambda + TA^2}{(R+1)^2\lambda}},
\end{aligned}
\tag{A.9}
$$

where the first inequality follows from Cauchy-Schwarz inequality, the second inequality follows from the definition of $\overline{\sigma}_t$ and Lemma C.2.

Substituting (A.8) and (A.9) into (A.7), we have

$$
I_1 = \widetilde{O}\left(C^2 d\sqrt{\sum_{t=1}^T \sigma_t^2} + C^2\sqrt{dT} + CR\sqrt{dT}\right).
\tag{A.10}
$$

By Lemma B.4,

$$
\begin{aligned}
I_2(\ell) &\le \mathbb{E}\left[\sum_{t=1}^T \mathbb{1}(f(t)=\ell)\min\left\{2, 2\gamma_{t,\ell}\|\mathbf{a}_{t,\ell}\|_{\mathbf{\Sigma}_{t,\ell}^{-1}}\right\}\right] \\
&\le 2\mathbb{E}\underbrace{\left[\sum_{t=1}^T \mathbb{1}(f(t)=\ell)\min\left\{1, \gamma_{t,\ell}\|\mathbf{a}_{t,\ell}\|_{\mathbf{\Sigma}_{t,\ell}^{-1}}\right\}\right]}_{I_4}
\end{aligned}
\tag{A.11}
$$

Again, we divide $[T]$ into two parts. Let $\mathcal{J}_1 = \{t \in [T] | \|\mathbf{a}_{t,\ell}/\overline{\sigma}_t\|_{\mathbf{\Sigma}_{t,\ell}^{-1}} > 1\}, \mathcal{J}_2 = \{t \in [T] | \|\mathbf{a}_{t,\ell}/\overline{\sigma}_t\|_{\mathbf{\Sigma}_{t,\ell}^{-1}} \le 1\}$.

$$
\begin{aligned}
\sum_{t\in\mathcal{J}_1}\mathbb{1}(f(t)=\ell)\min\left\{1, \gamma_{t,\ell}\|\mathbf{a}_{t,\ell}\|_{\mathbf{\Sigma}_{t,\ell}^{-1}}\right\} &\le \sum_{t\in\mathcal{J}_1}\mathbb{1}(f(t)=\ell)\cdot 1 \\
&\le \sum_{t=1}^T \mathbb{1}(f(t)=\ell)\min\left\{1, \|\mathbf{a}_{t,\ell}\|_{\mathbf{\Sigma}_{t,\ell}^{-1}}^2\right\} \\
&\le 2d\log\frac{(R+1)^2\lambda + TA^2}{(R+1)^2\lambda},
\end{aligned}
\tag{A.12}
$$

where the second inequality follows from the definition of $\mathcal{J}_1$, the second inequality holds due to Lemma C.2.

$$
\begin{aligned}
&\sum_{t\in\mathcal{J}_2}\mathbb{1}(f(t)=\ell)\min\left\{1, \gamma_{t,\ell}\|\mathbf{a}_{t,\ell}\|_{\mathbf{\Sigma}_{t,\ell}^{-1}}\right\} \\
&\le \gamma_{T,\ell}\sqrt{\sum_{t=1}^T \overline{\sigma}_t^2}\sqrt{\sum_{t\in\mathcal{J}_2}\min\left\{1, \|\mathbf{a}_t/\overline{\sigma}_t\|_{\mathbf{\Sigma}_{t,\ell}^{-1}}^2\right\}} \\
&\le \gamma_{T,\ell}\sqrt{(R+1)^2 T/d + \sum_{t=1}^T \sigma_t^2}\sqrt{2d\log\frac{(R+1)^2\lambda + TA^2}{(R+1)^2\lambda}},
\end{aligned}
\tag{A.13}
$$

where the first inequality follows from Cauchy-Schwarz inequality, the second inequality follows from the definition of $\overline{\sigma}_t$ and Lemma C.2.

Substituting (A.12) and (A.13) into (A.11), we have

$$I_2(\ell) \leq 4d \log \frac{(R+1)^2\lambda + TA^2}{(R+1)^2\lambda} + 2\gamma_{T,\ell}\sqrt{(R+1)^2T/d + \sum_{t=1}^{T}\sigma_t^2}\sqrt{2d \log \frac{(R+1)^2\lambda + TA^2}{(R+1)^2\lambda}} \tag{A.14}$$

$$I_2(\ell) = \widetilde{O}\left(R\sqrt{Td} + d\sqrt{\sum_{t=1}^{T}\sigma_t^2}\right). \tag{A.15}$$

Substituting (A.10) and (A.15) into (A.4), we have

$$\mathbf{Regret}(T) = \widetilde{O}\left(C^2 d\sqrt{\sum_{t=1}^{T}\sigma_t^2} + C^2\sqrt{dT} + CR\sqrt{dT}\right). $$

$\square$

## A.2    Proof of Theorem 5.6

*Proof of Theorem 5.6.*  First we decompose the regret as follows.

$$\mathbf{Regret}(T) = \mathbb{E}\left[\sum_{t=1}^{T}\left(\langle \mathbf{a}_t^*, \boldsymbol{\mu}^*\rangle - \langle \mathbf{a}_t, \boldsymbol{\mu}^*\rangle\right)\right]$$

$$\leq \frac{1}{\Delta}\underbrace{\mathbb{E}\left[\sum_{t=1}^{T}\mathbb{1}(f(t) \leq \ell^*)\left(\langle \mathbf{a}_t^*, \boldsymbol{\mu}^*\rangle - \langle \mathbf{a}_t, \boldsymbol{\mu}^*\rangle\right)^2\right]}_{I_1}$$

$$+ \sum_{\ell=\ell^*+1}^{\ell_{\max}}\frac{1}{\Delta}\underbrace{\mathbb{E}\left[\sum_{t=1}^{T}\mathbb{1}(f(t) = \ell)\left(\langle \mathbf{a}_t^*, \boldsymbol{\mu}^*\rangle - \langle \mathbf{a}_t, \boldsymbol{\mu}^*\rangle\right)^2\right]}_{I_2(\ell)}, \tag{A.16}$$

where the first equality holds due to the definition in (3.3), the last inequality follows from the fact that either $\langle \mathbf{a}_t^*, \boldsymbol{\mu}^*\rangle - \langle \mathbf{a}_t, \boldsymbol{\mu}^*\rangle = 0$ or $\overline{\Delta}_T \leq \langle \mathbf{a}_t^*, \boldsymbol{\mu}^*\rangle - \langle \mathbf{a}_t, \boldsymbol{\mu}^*\rangle$. To bound $I_1$, we have

$$I_1 \leq \mathbb{E}\left[\sum_{t=1}^{T}\mathbb{1}(f(t) \leq \ell^*)\min\left\{4, \left(2\beta_{t,\ell^*}\|\mathbf{a}_{t,\ell^*}\|_{\boldsymbol{\Sigma}_t^{-1}} + 2\beta_{t,\ell^*}\|\mathbf{a}_t\|_{\boldsymbol{\Sigma}_t^{-1}}\right)^2\right\}\right]$$

$$\leq 2^{\ell^*}\underbrace{\mathbb{E}\left[\sum_{t=1}^{T}\min\left\{4, 16\beta_{t,\ell^*}^2\|\mathbf{a}_t\|_{\boldsymbol{\Sigma}_t^{-1}}^2\right\}\right]}_{I_3}, \tag{A.17}$$

where the first inequality holds due to Lemma A.6 and the second inequality follows from a similar argument as (A.6). To further bound $I_3$, we decompose $[T]$ into two non-overlapping sets: $\mathcal{I}_1 = \{t \in [T]|\|\mathbf{a}_t/\overline{\sigma}_t\|_{\boldsymbol{\Sigma}_t^{-1}} > 1\}, \mathcal{I}_2 = \{t \in [T]|\|\mathbf{a}_t/\overline{\sigma}_t\|_{\boldsymbol{\Sigma}_t^{-1}} \leq 1\}$. For $\mathcal{I}_1$, we have

$$\sum_{t\in\mathcal{I}_1}\min\left\{4, 16\beta_{t,\ell^*}^2\|\mathbf{a}_t\|_{\boldsymbol{\Sigma}_t^{-1}}^2\right\} \leq 4|\mathcal{I}_1|$$

$$\leq 4\sum_{t=1}^{T}\min\left\{1, \|\mathbf{a}_t/\overline{\sigma}_t\|_{\boldsymbol{\Sigma}_t^{-1}}^2\right\}$$

$$\leq 8d \log \frac{(R+1)^2 \lambda + TA^2}{(R+1)^2 \lambda}, \tag{A.18}$$

where the third inequality holds due to Lemma C.2. For $\mathcal{I}_2$, we have

$$\sum_{t \in \mathcal{I}_2} \min \left\{ 4, 9\beta_{t,\ell^*}^2 \|\mathbf{a}_t\|_{\boldsymbol{\Sigma}_t^{-1}}^2 \right\} \leq 16\beta_{T,\ell^*}^2 \max_{t \in [T]} \overline{\sigma}_t^2 \sum_{t \in \mathcal{I}_2} \min \left\{ 1, \|\mathbf{a}_t/\overline{\sigma}_t\|_{\boldsymbol{\Sigma}_t^{-1}}^2 \right\}$$

$$\leq 32\beta_{T,\ell^*}^2 (\max_{t \in [T]} \sigma_t^2 + (R+1)^2/d) d \log \frac{(R+1)^2 \lambda + TA^2}{(R+1)^2 \lambda}, \tag{A.19}$$

where the first inequality follows from the definition of $\mathcal{I}_2$, the second inequality follows from Lemma C.2.

Substituting (A.18) and (A.19) into (A.17), we have

$$I_1 = \widetilde{O} \left( C^2 R^2 d + d^2 C^2 \max_{t \in [T]} \sigma_t^2 \right). \tag{A.20}$$

To bound $I_2(\ell)$, by Lemma A.7, we have

$$I_2(\ell) \leq \mathbb{E} \left[ \sum_{t=1}^{T} \mathbb{1}(f(t) = \ell) \min \left\{ 4, 4\gamma_{t,\ell}^2 \|\mathbf{a}_t\|_{\boldsymbol{\Sigma}_{t,\ell}^{-1}}^2 \right\} \right]$$

$$\leq 4\mathbb{E} \underbrace{\left[ \sum_{t=1}^{T} \mathbb{1}(f(t) = \ell) \min \left\{ 1, \gamma_{t,\ell}^2 \|\mathbf{a}_t\|_{\boldsymbol{\Sigma}_{t,\ell}^{-1}}^2 \right\} \right]}_{I_4}. \tag{A.21}$$

We divide $[T]$ into two parts to calculate $I_4$. Let $\mathcal{J}_1 = \{t \in [T] | \|\mathbf{a}_t/\overline{\sigma}_t\|_{\boldsymbol{\Sigma}_{t,\ell}^{-1}} > 1\}, \mathcal{J}_2 = \{t \in [T] | \|\mathbf{a}_t/\overline{\sigma}_t\|_{\boldsymbol{\Sigma}_{t,\ell}^{-1}} \leq 1\}$. For $\mathcal{J}_1$, we have

$$\sum_{t \in \mathcal{J}_1} \mathbb{1}(f(t) = \ell) \min \left\{ 1, \gamma_{t,\ell}^2 \|\mathbf{a}_t\|_{\boldsymbol{\Sigma}_{t,\ell}^{-1}}^2 \right\} \leq |\mathcal{J}_1|$$

$$\leq \sum_{t \in [T]} \min \left\{ 1, \|\mathbf{a}_t\|_{\boldsymbol{\Sigma}_{t,\ell}^{-1}}^2 \right\}$$

$$\leq 2d \log \frac{(R+1)^2 \lambda + TA^2}{(R+1)^2 \lambda}, \tag{A.22}$$

where the second inequality follows from the fact that $\mathcal{J}_1 \subseteq [T]$, the third inequality holds due to Lemma C.2. For $\mathcal{J}_2$, we have

$$\sum_{t \in \mathcal{J}_2} \mathbb{1}(f(t) = \ell) \min \left\{ 1, \gamma_{t,\ell}^2 \|\mathbf{a}_t\|_{\boldsymbol{\Sigma}_{t,\ell}^{-1}}^2 \right\} \leq \gamma_{T,\ell}^2 \max_{t \in [T]} \overline{\sigma}_t^2 \sum_{t \in \mathcal{J}_2} \|\mathbf{a}_t\|_{\boldsymbol{\Sigma}_{t,\ell}^{-1}}^2$$

$$\leq \gamma_{T,\ell}^2 (\max_{t \in [T]} \sigma_t^2 + (R+1)^2/d) \sum_{t \in [T]} \min \left\{ 1, \|\mathbf{a}\|_{\boldsymbol{\Sigma}_{t,\ell}^{-1}}^2 \right\}$$

$$\leq \gamma_{T,\ell}^2 (\max_{t \in [T]} \sigma_t^2 + (R+1)^2/d) 2d \log \frac{(R+1)^2 \lambda + TA^2}{(R+1)^2 \lambda}, \tag{A.23}$$

where the second inequality follows from the definition of $\overline{\sigma}_t$ and $\mathcal{J}_2$ and the third inequality holds due to Lemma C.2.

Substituting (A.22) and (A.23) into (A.21), we have

$$I_2(\ell) = \widetilde{O}(dR^2 + d^2 \max_{t \in [T]} \sigma_t^2). \tag{A.24}$$

Finally, substituting (A.24) and (A.20) into (A.16), we have

$$\mathbf{Regret}(T) = \frac{1}{\Delta}\widetilde{O}\left(C^2 R^2 d + d^2 C^2 \max_{t \in [T]} \sigma_t^2\right).$$

$\square$

# B Proof of Technical Lemmas in Section A

## B.1 Proof of Lemma A.1

*Proof.* Let $\mathcal{S}(t) = \{1 \le i \le t | y_i \ne y_i^{\text{stoch}}\}$, $\mathbf{b}_t^{\text{stoch}} = \sum_{i=1}^{t} y_i^{\text{stoch}} \mathbf{x}_i$ and $\boldsymbol{\mu}_t^{\text{stoch}} = \mathbf{Z}_t^{-1} \mathbf{b}_t^{\text{stoch}}$. By Lemma C.1, we have that with probability at least $1 - \delta$, $\|\boldsymbol{\mu}_t^{\text{stoch}} - \boldsymbol{\mu}^*\|_{\mathbf{Z}_t} \le \beta_t + \sqrt{\lambda}\|\boldsymbol{\mu}^*\|_2$ holds for all $t \ge 1$.

Also, we have

$$\|\boldsymbol{\mu}_t - \boldsymbol{\mu}_t^{\text{stoch}}\|_{\mathbf{Z}_t} = \|\mathbf{Z}_t^{-1}(\mathbf{b}_t - \mathbf{b}_t^{\text{stoch}})\|_{\mathbf{Z}_t}$$

$$\le \sum_{i=1}^{t} \|\mathbf{Z}_t^{-1}(y_i^{\text{stoch}} - y_i)\mathbf{x}_i\|_{\mathbf{Z}_t}$$

$$\le \sum_{i=1}^{t} |y_i^{\text{stoch}} - y_i| \cdot \|\mathbf{x}_i\|_{\mathbf{Z}_t^{-1}}$$

$$\le C(t).$$

where the first inequality holds due to the triangle inequality and the last inequality holds due to $\|\mathbf{x}_i\|_{\mathbf{Z}_t^{-1}} \le 1$.

Hence, we can obtain

$$\|\boldsymbol{\mu}_t - \boldsymbol{\mu}^*\|_{\mathbf{Z}_t} \le \|\boldsymbol{\mu}_t^{\text{stoch}} - \boldsymbol{\mu}^*\|_{\mathbf{Z}_t} + \|\boldsymbol{\mu}_t - \boldsymbol{\mu}_t^{\text{stoch}}\|_{\mathbf{Z}_t} \le \beta_t + C(t) + \sqrt{\lambda}\|\boldsymbol{\mu}^*\|_2.$$

$\square$

## B.2 Proof of Lemma A.2

**Lemma B.1** (Lemma 3.3, Lykouris et al. 2018). Define the corruption level for a level $\ell$:

$$\text{Corruption}_{t,\ell} = \sum_{i=1}^{t} \frac{\mathbb{1}(f(i) = \ell)}{R+1} \cdot \sup_{\mathbf{a} \in \mathcal{D}_i} |r_i(\mathbf{a}) - r_i'(\mathbf{a})|.$$

Then we have for all $\ell \ge \ell^*$, with probability at least $1 - \delta$:

$$\text{Corruption}_{t,\ell} \le \log(1/\delta) + 3, \qquad \forall t \ge 1.$$

*Proof of Lemma A.2.* Applying Lemma B.1, we have for all $\ell \ge \ell^*$, with probability at least $1 - \delta/(2\ell^2)$: $\text{Corruption}_{t,\ell} \le \log(2\ell^2/\delta) + 3, \forall t \ge 1$.

Using a union bound over all $\ell \ge \ell^*$, we can prove the lemma. $\square$

## B.3 Proof of Lemma A.4

To prove the lemma, we first define the following two events:

$$\mathcal{E}_2 := \{\forall \ell \ge \ell^* \text{ and } t \ge 1, \|\boldsymbol{\mu}^* - \boldsymbol{\mu}_t\|_{\boldsymbol{\Sigma}_t} \le \beta_{t,\ell}\} \tag{B.1}$$

$$\mathcal{E}_3 := \{\forall \ell \ge \ell^* \text{ and } t \ge 1, \|\boldsymbol{\mu}^* - \boldsymbol{\mu}_{t,\ell}\|_{\boldsymbol{\Sigma}_{t,\ell}} \le \gamma_{t,\ell}\} \tag{B.2}$$

**Lemma B.2.** Let $\mathcal{E}_2$ be defined in (B.1). For any $0 < \delta < 1$, we have $\mathbb{P}(\mathcal{E}_2) \ge 1 - \delta$.

*Proof.* Applying Lemma A.1, we have that $\|\boldsymbol{\mu}_t - \boldsymbol{\mu}^*\|_{\boldsymbol{\Sigma}_t} \leq 8\sqrt{d\log\frac{(R+1)^2\lambda+tA^2}{(R+1)^2\lambda}\log(4t^2/\delta)} + 4\sqrt{d}\log(4t^2/\delta) + C\sqrt{d} + \sqrt{\lambda}\|\boldsymbol{\mu}^*\|_2$ for all $t \geq 1$ with probability at least $1 - \delta$. Note that $2^\ell \geq C$ for all $\ell \geq \ell^*$, which indicates that $\mathcal{E}_2$ occurs with probability at least $1 - \delta$. $\square$

**Lemma B.3.** *Let $\mathcal{E}_3$ be defined in (B.2). For any $0 < \delta < 1$, we have $\mathbb{P}(\mathcal{E}_3) \geq 1 - 2\delta$.*

*Proof.* Applying Lemma A.1, we have that $\|\boldsymbol{\mu}_{t,\ell} - \boldsymbol{\mu}^*\|_{\boldsymbol{\Sigma}_t} \leq 8\sqrt{d\log\frac{(R+1)^2\lambda+tA^2}{(R+1)^2\lambda}\log(4t^2T/\delta)} + 4\sqrt{d}\log(4t^2T/\delta) + \mathrm{Corruption}_{t,\ell}\sqrt{d} + \sqrt{\lambda}\|\boldsymbol{\mu}^*\|_2$ for all $t \geq 1$ with probability at least $1 - \delta/\ell$. Here we use the fact that $\ell \leq T$. Applying Lemma A.2 and a union bound, we have $\mathcal{E}_3$ occurs with probability at least $1 - 2\delta$. $\square$

*Proof of Lemma A.4.* This lemma can be proved by a union bound on $\mathcal{E}_2$ and $\mathcal{E}_3$ with Lemmas B.2 and B.3. $\square$

## B.4 Proof of Lemma A.6

*Proof.* For simplicity, let $\mathcal{A}_{t,\ell} = \{\boldsymbol{\mu}|\|\boldsymbol{\mu} - \boldsymbol{\mu}_t\|_{\boldsymbol{\Sigma}_t} \leq \beta_{t,\ell}\}$, $\mathcal{B}_{t,\ell} = \{\boldsymbol{\mu}|\|\boldsymbol{\mu} - \boldsymbol{\mu}_{t,\ell}\|_{\boldsymbol{\Sigma}_{t,\ell}} \leq \gamma_{t,\ell}\}$. Let $\boldsymbol{\mu}_t^m = \mathrm{argmax}_{\boldsymbol{\mu}\in\mathcal{C}_{t,f(t)}}\langle \mathbf{a}_t, \boldsymbol{\mu}\rangle$. Then we have

$$
\begin{aligned}
\langle \mathbf{a}_t, \boldsymbol{\mu}^*\rangle &\geq \langle \mathbf{a}_t, \boldsymbol{\mu}_t\rangle - \beta_{t,\ell^*}\|\mathbf{a}_t\|_{\boldsymbol{\Sigma}_t^{-1}} \\
&\geq \langle \mathbf{a}_t, \boldsymbol{\mu}_t^m\rangle - 2\beta_{t,\ell^*}\|\mathbf{a}_t\|_{\boldsymbol{\Sigma}_t^{-1}} \\
&\geq \langle \mathbf{a}_{t,\ell^*}, \boldsymbol{\mu}_t^m\rangle - 2\beta_{t,\ell^*}\|\mathbf{a}_t\|_{\boldsymbol{\Sigma}_t^{-1}} \\
&\geq \langle \mathbf{a}_{t,\ell^*}, \boldsymbol{\mu}_t\rangle - \beta_{t,\ell^*}\|\mathbf{a}_{t,\ell^*}\|_{\boldsymbol{\Sigma}_t^{-1}} - 2\beta_{t,\ell^*}\|\mathbf{a}_t\|_{\boldsymbol{\Sigma}_t^{-1}} \\
&\geq \max_{\boldsymbol{\mu}\in\mathcal{A}_{t,\ell^*}}\langle \mathbf{a}_{t,\ell^*}, \boldsymbol{\mu}\rangle - 2\beta_{t,\ell^*}\|\mathbf{a}_{t,\ell^*}\|_{\boldsymbol{\Sigma}_t^{-1}} - 2\beta_{t,\ell^*}\|\mathbf{a}_t\|_{\boldsymbol{\Sigma}_t^{-1}}, \\
&\geq \max_{\boldsymbol{\mu}\in\mathcal{C}_{t,\ell^*}}\langle \mathbf{a}_{t,\ell^*}, \boldsymbol{\mu}\rangle - 2\beta_{t,\ell^*}\|\mathbf{a}_{t,\ell^*}\|_{\boldsymbol{\Sigma}_t^{-1}} - 2\beta_{t,\ell^*}\|\mathbf{a}_t\|_{\boldsymbol{\Sigma}_t^{-1}}, \quad (B.3)
\end{aligned}
$$

where the first inequality holds since $\boldsymbol{\mu}^* \in \mathcal{C}_{t,\ell^*} \subseteq \mathcal{A}_{t,l^*}$, the second inequality holds since $\boldsymbol{\mu}_t^m \in \mathcal{C}_{t,f(t)} \subseteq \mathcal{A}_{t,l^*}$, the third inequality holds by the definition of $\mathbf{a}_t$ and $\boldsymbol{\mu}_t^m$, the fourth inequality holds since $\boldsymbol{\mu}_t^m \in \mathcal{A}_{t,\ell^*}$, the fifth inequality holds since $\boldsymbol{\mu}_t \in \mathcal{A}_{t,\ell^*}$, the last one holds since $\mathcal{C}_{t,\ell^*} \subseteq \mathcal{A}_{t,\ell^*}$. By the definition of $\mathcal{E}_1$ and $\mathbf{a}_{t,\ell^*}$, we have

$$
\max_{\boldsymbol{\mu}\in\mathcal{C}_{t,\ell^*}}\langle \mathbf{a}_{t,\ell^*}, \boldsymbol{\mu}\rangle = \max_{\mathbf{a}\in\mathcal{D}_t}\max_{\boldsymbol{\mu}\in\mathcal{C}_{t,\ell^*}}\langle \mathbf{a}, \boldsymbol{\mu}\rangle \geq \max_{\mathbf{a}\in\mathcal{D}_t}\langle \mathbf{a}, \boldsymbol{\mu}^*\rangle = \langle \mathbf{a}_t^*, \boldsymbol{\mu}^*\rangle. \quad (B.4)
$$

Combining (B.3) with (B.4), we have $\langle \mathbf{a}_t^* - \mathbf{a}_t, \boldsymbol{\mu}^*\rangle \leq 2\beta_{t,\ell^*}\|\mathbf{a}_t\|_{\boldsymbol{\Sigma}_t^{-1}} + 2\beta_{t,\ell^*}\|\mathbf{a}_{t,\ell^*}\|_{\boldsymbol{\Sigma}_t^{-1}}$. $\square$

## B.5 Proof of Lemma A.7

*Proof.* We have

$$
\langle \mathbf{a}_t^* - \mathbf{a}_t, \boldsymbol{\mu}^*\rangle \leq \max_{\boldsymbol{\mu}\in\mathcal{C}_{t,\ell}}\langle \mathbf{a}_t, \boldsymbol{\mu}\rangle - \langle \mathbf{a}_t, \boldsymbol{\mu}^*\rangle \leq 2\gamma_{t,\ell}\|\mathbf{a}_t\|_{\boldsymbol{\Sigma}_{t,\ell}^{-1}},
$$

where the first inequality follows from the fact that $\boldsymbol{\mu}^* \in \mathcal{C}_{t,\ell}$ and the definition of $\mathbf{a}_t$, the second inequality holds since $\boldsymbol{\mu}^* \in \mathcal{C}_{t,\ell}$ on the event $\mathcal{E}_1$. $\square$

# C Auxiliary Lemmas

**Lemma C.1** (Theorem 4.1, Zhou et al. 2020)**.** *Let $\{\mathcal{G}_t\}_{t=1}^\infty$ be a filtration, $\{\mathbf{x}_t, \eta_t\}_{t\geq 1}$ a stochastic process so that $\mathbf{x}_t \in \mathbb{R}^d$ is $\mathcal{G}_t$-measurable and $\eta_t \in \mathbb{R}$ is $\mathcal{G}_{t+1}$-measurable. Fix $R, L, \sigma, \lambda > 0$, $\boldsymbol{\mu}^* \in \mathbb{R}^d$. For $t \geq 1$ let $y_t = \langle \boldsymbol{\mu}^*, \mathbf{x}_t\rangle + \eta_t$ and suppose that $\eta_t, \mathbf{x}_t$ also satisfy*

$$
|\eta_t| \leq R, \mathbb{E}[\eta_t|\mathcal{G}_t] = 0, \mathbb{E}[\eta_t^2|\mathcal{G}_t] \leq \sigma^2, \|\mathbf{x}_t\|_2 \leq L.
$$

*Then, for any $0 < \delta < 1$, with probability at least $1 - \delta$ we have $\forall t > 0$,*

$$
\|\boldsymbol{\mu}_t - \boldsymbol{\mu}^*\|_{\mathbf{Z}_t} \leq \beta_t + \sqrt{\lambda}\|\boldsymbol{\mu}^*\|_2,
$$

where for $t \geq 1$, $\boldsymbol{\mu}_t = \mathbf{Z}_t^{-1}\mathbf{b}_t$, $\mathbf{Z}_t = \lambda\mathbf{I} + \sum_{i=1}^{t}\mathbf{x}_i\mathbf{x}_i^{\top}$, $\mathbf{b}_t = \sum_{i=1}^{t}y_i\mathbf{x}_i$, and

$$\beta_t = 8\sigma\sqrt{d\log\frac{d\lambda + tL^2}{d\lambda}\log(4t^2/\delta)} + 4R\log(4t^2/\delta).$$

**Lemma C.2** (Lemma 11, Abbasi-Yadkori et al. 2011). *For any $\lambda > 0$ and sequence $\{\mathbf{x}_t\}_{t=1}^{T} \subset \mathbb{R}^d$ for $t \in 0 \cup [T]$, define $\mathbf{Z}_t = \lambda\mathbf{I} + \sum_{i=1}^{t}\mathbf{x}_i\mathbf{x}_i^{\top}$. Then, provided that $\|\mathbf{x}_t\|_2 \leq L$ holds for all $t \in [T]$, we have*

$$\sum_{t=1}^{T}\min\{1, \|\mathbf{x}_t\|_{\mathbf{Z}_{t-1}^{-1}}^{2}\} \leq 2d\log\frac{d\lambda + TL^2}{d\lambda}.$$

## D  Experiments

In this section, we conduct experiments and evaluate the performance our algorithm Multi-level OFUL, along with the baselines, OFUL (Abbasi-Yadkori et al., 2011), weighted OFUL (Zhou et al., 2020) and the greedy algorithm proposed by Bogunovic et al. (2021) under different corruption levels. We repeat each baseline algorithm for 10 times and plot their regrets w.r.t. number of rounds in Figure 1.

### D.1  Experimental Setup

Following Bogunovic et al. (2021), we let the adversary always corrupt the first $k$ rounds, and leave the rest $T - k$ rounds intact. According to our definition in (3.2), our design can simulate the cases where corruption level is $2k$.

**Model parameters.** Recall that corrupted linear contextual bandits defined in Section 3, we consider $B = 1$, $A = 1$ $d = 20$ and $R = 0.5$ and fix $\boldsymbol{\mu}^*$ as $\left(\frac{1}{\sqrt{d}}, \cdots, \frac{1}{\sqrt{d}}\right)^{\top}$. We set $\sigma_t$ as a random variable which is independently and uniformly chosen from $[0, 0.05]$ in each round $t$. Note that $\langle\mathbf{a}, \boldsymbol{\mu}^*\rangle \in [-1, 1]$ always hold for any eligible $\mathbf{a}$ under our setting of parameters.

**Attack method.** In the first $k$ rounds, the adversary always trick the learner by flipping the value of $\boldsymbol{\mu}^*$, i.e., $r_t(\mathbf{a}) = -\langle\mathbf{a}, \boldsymbol{\mu}^*\rangle + \epsilon_t(\mathbf{a})$ for all $t \in [k]$ and $\mathbf{a} \in \mathcal{D}_t$.

**Decision set.** We consider $|\mathcal{D}_t| = 20$ for all $t \geq 1$. In each of the first $k$ rounds, we generate the 20 actions in $\mathcal{D}_t$ independently, each having entries drawn i.i.d. from the uniform distribution on $\left[-\frac{1}{\sqrt{d}}, \frac{1}{\sqrt{d}}\right]$. For the following uncorrupted rounds, however, we use a fixed $\mathcal{D}$ generated in the same way.

Intuitively, non-robust algorithm will "learn" the flipped $\boldsymbol{\mu}^*$ faster with diversified action vectors. As a result, the learner is likely to select the same nonoptimal action for a huge number of rounds afterwards, making it even more difficult to learn the true $\boldsymbol{\mu}^*$.

**Noise synthesis.** We generate identical noises $\epsilon_t$ for all $\mathbf{a} \in \mathcal{D}_t$ at each round $t$, i.e., $\epsilon_t(\mathbf{a}) = \epsilon_t$. To generate $\epsilon_t$, we first generate $\epsilon_t'$ subject to $\mathcal{N}(0, \sigma_t^2)$ and let

$$\epsilon_t = \begin{cases} -R, & \epsilon_t' < -R \\ R, & \epsilon_t' > R \\ \epsilon_t', & \text{otherwise} \end{cases}.$$

 **D.2   Results and Discussion**

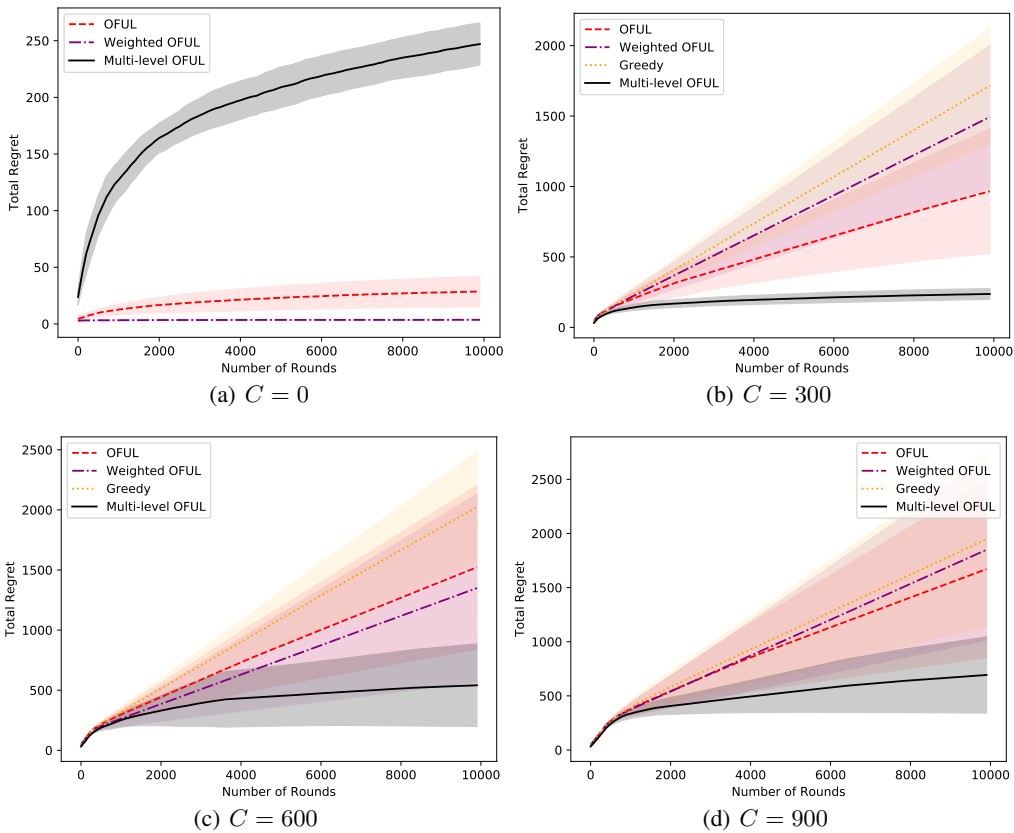

Figure 1: Regret plot against number of rounds under corruption level from 0 to 900 averaged in 10 trials.

We plot the regret with respect to the number of rounds in Figure 1. The results are averaged over 10 trials. In the setting where $C = 0$ (Figure 1(a)), we only plot the regret of OFUL, weighted OFUL and Multi-level OFUL, and do not plot the regret of the greedy algorithm since its regret is much worse than the other three algorithms.

We have the following observations from Figure 1. For the corruption-free case $C = 0$ (Figure 1(a)), our proposed Multi-level OFUL behaves worse than weighted OFUL and OFUL, which is not surprising since Multi-level OFUL has additional algorithm design to deal with the corruption and it may pay additional price in regret in the absence of corruption. Weighted OFUL outperforms OFUL remarkably since it takes advantage of the information concerning the variance of noise. For the corruption case (Figure 1(b) to 1(d)), our Multi-level OFUL outperforms other baseline algorithms by a large margin, which suggests that it can deal with the corruption successfully.