# OpenReview forum: "Linear Contextual Bandits with Adversarial Corruptions"
_NeurIPS.cc/2021/Conference — NeurIPS 2021 Submitted_

### Official Review · Reviewer_N6Ux · 2021-07-15

**Rating:** 5
**Confidence:** 4

**Summary:**

This paper proposes an algorithm that achieves the first corruption-robust result for contextual linear stochastic bandits with infinite arms using the idea of multi-level sampling rates. The regret bound has multiplicative dependence on C^2 and is variance-aware. The algorithm recovers the optimal regret in terms of T and gap when no corruptions are presented.

**Limitations And Societal Impact:**

The authors have clearly stated the theoretical limitation of their works.

**Main Review:**

--Originality:
The main idea to achieve robustness, as the authors stated, is from Lykouris et al. (2018), where the algorithm assigns multi-level sampling rates and therefore remains multi-level confidence sets. One major concern to me is that, although the authors claim in Remark 4.1 that, instead of doing arm elimination as in Lykouris et al. (2018), they are using UCB-type (OFUL) algorithm and maintain multi-level estimators, I find such idea is similar to Lykouris et al. (2020).  Lykouris et al. (2020) also combine UCB-type methods with the multi-layer sampling scheme and maintain different base learners for each layer. Because Lykouris et al. (2020) are considering linear MDP and here the authors are considering contextual linear bandits with changing decision set, so these two results are not directly comparable. But due to these previous works, I think the technical novelty of this paper is relatively limited.
Lykouris et al. (2018): Stochastic bandits robust to adversarial corruption
Lykouris et al. (2020): Corruption robust exploration in episodic reinforcement learning.

--Quality:
I think all the claims are well supported but this C^2 multiplicative dependence seems not good enough. It would be better if authors can provide some lower bound or at least some intuition on this C^2 multiplicative term.

I also have some suggestions on the experiment although this is not the main part of this paper: Both OFUL and weighted OFUL are deterministic algorithms, and the Greedy are designed for a stronger adversary where it can decide corruption after observing the learner’s action. So I am not surprised that Multi-level OFUL can outperform these three. Could you compare Multi-level OFUL with some randomized algorithms, for example, exp3?

--Clarity:
This paper is clearly written overall but the proof sketch is a little confusing: At the first glance I am not sure where the C^2 comes from, then I realize it comes from $C\beta_{t,l}$ where $\beta_{t,l}$ contains another C. But at the first glance I saw another $\beta_t$ in line 246, which makes me a little confused. I would like to suggest the author give some intuitive explanation on why getting this extra C dependence compared to Lykouris et al. (2018). Is there any specialty in contextual linear bandits analysis that results in this extra C?

--Significance
This is the first corruption-robust result in contextual linear bandits with infinite arms, so the result is non-trivial. Nevertheless, I think it is not good enough and the technical novelty is limited.


**Time Spent Reviewing:**

3.5

---

> ### Author Response · Authors · 2021-08-10
> **Response to Reviewer N6Ux**
>
> Q. “I think all the claims are well supported but this $C^2$ multiplicative dependence seems not good enough..”
>
> A.
> We agree that the $C^2$ dependence is not ideal. Yet our work is the first attempt to provide a linear contextual bandit algorithm that is robust to corruption. We believe our contribution is significant and we will study how to reduce $C^2$ to $C$ in future work.
>
> Q. “It would be better if authors can provide some lower bound or at least some intuition on this $C^2$ multiplicative term”
>
> A.
> Actually, $\beta_t$ in line 246 does not share the same meaning with $\beta_{t, \ell}$ in the algorithm. We will make it unambiguous in the revision.
> The $C^2$ dependence is due to the factor $C$ in $\beta_{t, \ell}$ and the summation of bonus terms across different levels. (See the details in the proof (Line 424-428). )
>
> Q. “Could you compare Multi-level OFUL with some randomized algorithms, for example, exp3?”
>
> A.
> Yes, we have added the comparison with the exp3 baseline. We reported the results in the following table $(C=300, 600)$.
> ________________________________________________
>
>
> $C = 300. $
>
> Algorithm         |    t=5000    |    t=8000   |    t=10000
> -|-|-|-
> Multi-level OFUL  |     246      |     270     |      285
> Weighted OFUL     |     777      |    1205     |      1491
> OFUL              |     694      |    1023     |      1183
> EXP3              |     510      |     706     |      828
> GREEDY            |     1076    |     1689    |      2097
> ---------------------------------------------------------------
>
>
>
> $C = 600. $
>
> Algorithm         |    t=5000    |    t=8000   |    t=10000
> -|-|-|-
> Multi-level OFUL  |     339      |     376     |      393
> Weighted OFUL     |     769      |    1147     |      1398
> OFUL              |     787      |    1102     |      1312
> EXP3              |     680      |     887     |      990
> GREEDY            |     1057     |     1621    |      1997

---

### Official Review · Reviewer_9fU4 · 2021-07-15

**Rating:** 4
**Confidence:** 4

**Summary:**

The authors study  a linear contextual bandit problem in the presence of adversarial corruptions that are bounded by an unknown C.  Specifically, they propose an algorithm that uses a parallel-learning idea from Lykouris et. al, 2019 using multi-level partition of the observed data.  To deal with varying variance in the observations, they use a ridge regression estimator similar to the earlier work of Kirschner and Krause, 2018; and Zhou et al., 2020.   Finally, they show their algorithm produces a regret that is quadratic in C and the dimensionality of the context vectors.

**Ethical Concerns:**

None.

**Limitations And Societal Impact:**

Modeling the corruptions realistically is an important aspect of such studies.  I am not completely bought into the proposed model of corruptions in this paper.  An adversary in general does not have the power to corrupt all observations.

**Main Review:**

The algorithm proposed in this paper borrow heavily from earlier works - Lykouris et. al, 2019 for the parallel-learning idea using partitioning of the observed data; Kirschner and Krause, 2018; and Zhou et al., 2020 for dealing with variance in the observed data; and almost mirrors the results of Lee et. al., 2021.  In fact, when C is large, their result can be seen to be underperforming compared to the result in Lee et. al, 2021.

How does this model compare to others when the corruption is unbounded but only corrupts \eps fraction of the observations and stochastic in the remaining observations?  The assumption of a bounded C seems a bit strong.  Further, does one need all the parallel-learning machinery to estimate C?

**Time Spent Reviewing:**

1.5

---

> ### Author Response · Authors · 2021-08-10
> **Response to Reviewer 9fU4**
>
> Q. “How does this model compare to others when the corruption is unbounded but only corrupts \eps fraction of the observations and stochastic in the remaining observations? ”
>
> A.
> When the corruption is bounded, our corruption model admits the $\epsilon$ contamination model (i.e., $\epsilon$ fraction of the observations are corrupted), with $C = \epsilon\cdot T$. When the corruption is allowed to be unbounded, then it is a drastically different model that may need a robust statistical estimator. This is beyond the scope of this work.
>
> Q. “The assumption of a bounded $C$ seems a bit strong. ”
>
> A.
>  Corruption model with bounded $C$ is a very standard and widely studied model in online learning and bandit literature. Bounded $C$ might be a bit of a strong assumption, yet we choose to study this corruption model in order to better position our result in the literature.
>
> Q. “Further, does one need all the parallel-learning machinery to estimate $C$?”
>
> A.
> Our algorithm needs the parallel learning scheme to estimate $C$. We don’t know how to estimate $C$ if a paralell-learning scheme is not used.

---

### Official Review · Reviewer_3jZ9 · 2021-07-16

**Rating:** 5
**Confidence:** 4

**Summary:**

This paper studies the linear contextual bandit problem with adversarial noises in the reward function. The authors assume that the decision set can change over time; that reward can be corrupted up to a corruption level (similar to previous work); and that an upper-bound on the true variance of the reward is revealed after each instance. An algorithm, which is based on the linear bandit algorithm OFUL with two notable modifications, is proposed for the problem. First, it considers multiple levels of possible corruption noise to deal with unknown value of C and second it considers a weighted version of the ridge regression estimator to deal with varying variances (this modification was already present in previous work). The authors prove a regret upper-bound for their algorithm and compare it to the literature. They also provide some experiments in the supplementary material.

**Limitations And Societal Impact:**

The authors did not mention the limitations and potential negative societal impact of their work.

**Main Review:**

The paper considers an existing problem: linear contextual bandit problem with adversarial noises (li et al, 2019; Bogunovic et al. 2021; Lee et al. 2021). The main contribution is to incorporate the possibility to deal with varying decision sets, together with varying variances and an unknown corruption level. The upper-bounds have no dependence on the number of actions but are linear in the dimension d which is also common for linear contextual bandits.


On the other hand, the paper suffers in my opinion the following weaknesses:
	1. The paper appears to be limited to a combination of existing techniques: adaptation to an unknown level of corruption (Lykouris et al., 2018); varying variances treated with a weighted version of OFUL (Zhou et al., 2021); variable decision sets (standard in contextual linear bandits). The fact that these results can be combined together is not surprising, and thus the contribution could be considered incremental.
	2. The regret bounds seem sub-optimal in the level of corruption (they are of order C^2 when existing bounds seem to be of order C). The authors should discuss more this sub-optimality, is it due to the unknown corruption level? Why should we incur it?

Other comments:
--------------
3. The authors state that if C is known and the variances are fixed then one could directly apply OFUL with a modified variance (to add the corruption). This yields several questions:
	a. The regret bound would then be of order O((R+C)d\sqrt{T}), right? It would be informative to write it, so that we can compare it with the bound of Thm. 5.1.
	b. It is then claimed that one of the problems is the varying variances. But this problem was already solved by Weighed OFUL (Thm. 4.2 of Zhou et al 2021) for OFUL without corruption. Isn't it possible to apply the same reasoning with this algorithm?
	c. For the adaptation to the unknown value of C, I am wondering whether it is not possible to just apply the Corral algorithm (see Cor. 6 of [1]) to (Weighed)-OFUL with an exponential grid of possible values for C (from 1 to T). Wouldn't this imply a bound of order C \sqrt{T} log T?

4. The assumption that the variance is revealed by the adversary (l. 135) is not clear and should be better motivated. We understand that it was already done by Kirschner and Krause (2018) and Zhou et al (2021) but examples of practical applications would be enjoyable to make the paper more self-contained. Similarly, the assumption of varying decision sets is standard in linear bandits but a few lines to recall why this allows dealing with contexts could be helpful for a reader new to the area.


5. About the experiments:
	a.The results seem significantly different when C = 0 and 300. What is the intermediate regime? For what level of corruption does Multi-level OFUL outperform algorithms that do not consider corruption?
	b. I regret that the algorithm is only compared to baselines that are not designed to deal with corruption and suffer linear regrets. It would be interesting to compare Multi-level OFUL with algorithms for linear bandits with corruptions. This could be done by considering fixed variance and fixed decision sets for instance to apply existing algorithms, so that we can see the actual cost of having a more general algorithm. The algorithm could also be compared with the version of OFUL which knows C and takes into account the corruption.


Minor remarks:
-------------
- How a \min in (6.3) is obtained using lemma 6.5 is not clear and should be clarified. Same for the \min in (6.8) using lemma 6.6?
- How substituting (6.5) and (6.6) into (6.3) gives (6.7) should also be more detailed.



[1] Agarwal et al. Corralling a Band of Bandit Algorithms, 2017.

**Time Spent Reviewing:**

3

---

> ### Author Response · Authors · 2021-08-10
> **Response to Reviewer 3jZ9**
>
> Q. “The authors state that if C is known and the variances are fixed then one could directly apply OFUL with a modified variance (to add the corruption). This yields several questions: a. The regret bound would then be of order $O((R+C)d\sqrt{T})$, right?”
>
> A. Yes, it is correct. We will add this statement in the revision.
>
> Q. “For the adaptation to the unknown value of $C$, I am wondering whether it is not possible to just apply the Corral algorithm (see Cor. 6 of [1]) to (Weighed)-OFUL with an exponential grid of possible values for $C$ (from 1 to $T$). Wouldn't this imply a bound of order $C \sqrt{T} log T$?”
>
> A.
> Thank you for the suggestion. It seems that the Corral algorithm is not directly applicable to OFUL (or LinUCB) because OFUL (or LinUCB) does not satisfy the stability condition required by the analysis of the Corral algorithm.
>
>
> Q. “The assumption that the variance is revealed by the adversary (l. 135) is not clear and should be better motivated. Similarly, the assumption of varying decision sets is standard in linear bandits but a few lines to recall why this allows dealing with contexts could be helpful for a reader new to the area.”
>
> A. Thank you for the advice. We will introduce the motivation behind these assumptions in the revision. In practical applications, the environment may change over time, and that's the reason why heteroscedastic noise needs to be considered. Also, on the theoretical aspect, the setting of heteroscedastic noise here can be extended to the MDP case, where the "noise" is causing by the transition of states and the variance can be estimated by the agent.
>
> ______________________________________________
>
> Add experiment when $C = 60, 120, 180$.
>
>
> $C = 60$.
>
> |Algo.|t=3000|t=4000|t=5000|
> |-|-|-|-|
> |Multi-level OFUL|197|213|224|
> |Weighted OFUL     |     313      |     413     |      514|
> |OFUL              |     129      |     150     |      162|
> |GREEDY            |     480      |     637     |      795|
>
> $C = 120$
>
> Algorithm         |    t=3000    |    t=4000   |    t=5000
> -|-|-|-
> Multi-level OFUL  |     203      |     218     |      230
> Weighted OFUL     |     423      |     551     |      679
> OFUL              |     276      |     345     |      413
> GREEDY            |     570      |     753     |      936
>
>
>
>
>
> $C = 180$.
>
> Algorithm         |    t=3000    |    t=4000   |    t=5000
> -|-|-|-
> Multi-level OFUL  |     219      |     232     |      243
> Weighted OFUL     |     450      |     587     |      724
> OFUL              |     425      |     534     |      644
> GREEDY            |     567      |     744     |      922
>
>
>
> Add experiment for WOFUL(Weighted OFUL) when $C$ is known.
> $C = 300$ (Weighted OFUL with Known $C$)
>
> Algorithm         |    t=3000    |    t=4000   |    t=5000
> -|-|-|-
> Multi-level OFUL  |     232      |     258     |      281
> Weighted OFUL     |     565      |     735     |      905
> OFUL              |     559      |     711     |      862
> WOFUL (Known C)   |     240      |     252     |      261
> GREEDY            |     652      |     853     |      1054

---

> > ### Comment · Reviewer_3jZ9 · 2021-08-19
> > **Quick response to the authors**
> >
> > I thank the authors for their response to my review. While I am currently travelling (for the next 1.5 week) and in a very tight time schedule,  I only give a high level response. I also read other reviews.
> >
> > I am not fully satisfied with the answers from the authors. Why does OFUL does not satisfy the stability condidtion from Coral and isn't is possible to fix it? But I decide to keep my score of 5 because I still believe that the contribution is mostly limited to a combination of existing works.

---

> > > ### Author Response · Authors · 2021-08-23
> > > **Reply to quick response**
> > >
> > > Thank you for your quick response, in particular, for telling us that you’re not satisfied with our answer to the application of the CORRAL algorithm. Let us explain here in detail why Corral is not directly applicable in our problem setting.
> > >
> > > The stability condition introduced in Agarwal et al.[1] requires that the base algorithm adapt to the value of $\rho$ (See Definition ) which is unknown. However, OFUL needs to know $\rho$ to compute the range of noise and construct a confidence set.
> > >
> > > We noticed that Pacchiano et al. [2] modified CORRAL so that it can be compatible with LinUCB. This will yield a very different algorithm from ours and a more careful analysis is needed to verify this idea. Moreover, we also prove a gap-dependent regret bound for our algorithm. However, it is not clear if we can also derive a gap-dependent regret bound for the Corral-based algorithm.
> > >
> > >
> > > [1] Agarwal et al. Corralling a Band of Bandit Algorithms, 2017.
> > >
> > > [2] Pacchiano et al. Model Selection in Contextual Stochastic Bandit
> > > Problems, 2020.

---

### Official Review · Reviewer_5Lqk · 2021-07-18

**Rating:** 6
**Confidence:** 4

**Summary:**

The paper studies the problem of linear contextual bandits where the rewards of the actions are not always stochastic but rather they are corrupted in C of the T rounds. This model of adversarial corruptions is well understood in the multi-armed bandit setting. The paper combines the OFUL algorithm that works for linear contextual bandits without corruptions with a multi-level algorithm that is robust to corruptions for the multi-armed bandit setting. Unlike multi-armed bandits, in linear contextual bandits, the action space is not finite and therefore techniques that are based on eliminating actions after selecting them enough times are not as direct. Moreover, the variance of the estimator can be changing over time which is dealt with via the use of a weighted ridge regression estimator. The resulting guarantee of Theorem 5.1 matches the regret bound of OFUL when C is constant and adapts to the corruption level C.


**Limitations And Societal Impact:**

I do not see a negative societal impact of the work. As explained above, I would like to have more clarity with respect to technical limitations of the work and I am looking forward to their response.

**Main Review:**

Overall I enjoyed reading the paper as it provides a nice extension of the multi-level algorithm to the linear contextual bandit setting and tackles the additional complications. The paper is well written, modularly explained, and, to the best of my understanding, correct.

On the other hand, there are a few places that would be worth clarifying.

1. My first question is with respect to the comparison to the work of Lee et al. (2021) which has a stronger regret guarantee. The authors distinguish their work via mentioning that the latter paper has a stronger assumption. Is there a reason why the algorithm of Lee et al. (2021) would not extend to the setting studied in this paper?

2. My second question is with respect to the variance $\sigma_t$. It seems that  the paper does not allow the variance to also be corrupted and it also requires $\sigma_t$ to be observable. It would be great if these assumptions are necessary or if they are for convenience of exposition. A related question: Do we know anything about the case where $\sigma_t$ is not observable in the uncorrupted case?

3. The claim that the non-finiteness of the version space means that elimination techniques cannot be applied is not really accurate. There are settings such as dynamic pricing where elimination-based application of a similar multi-level technique provide robustness to adversarial corruptions. For example, here are two references:

-- Robust Dynamic Pricing with Demand Learning in the Presence of Outlier Customers (Chen and Wang, 2020)

-- Contextual Search in the Presence of Irrational Agents (Krishnamurthy, Lykouris, Podimata, Schapire, STOC 2021)

It would be useful if the paper pinpoints what is the feature that exists in linear contextual bandits that requires to move beyond elimination-based techniques.

4. Extending the above point, I have a concern that, on a technical level, the paper combines three existing techniques (multi-level technique for MAB with corruptions, OFUL for linear contextual bandits, and weighted ridge regression for varying variance). It is not clear to me whether there exists some insight that is really required when these techniques need to coexist. It would be great if the authors could explain whether there is some roadblock that arises in the intersection of these techniques that does not exist in each of them independently.

Despite the above concerns, I am generally positive about the paper and would be happy to see the paper at NeurIPS. I encourage the authors to respond to the above comments and I am looking forward to the author rebuttal.


Minor comments and typos:
-- In line 11, probably you mean "Lee et al. (2021)" instead of "Li et al. (2019)".
-- In line 137, I think it would be cleaner to say that $\mu^*$ is selected arbitarily in the beginning rather than "observed by the adversary"; in the protocol, the adversary (or nature) selects $\mu*$ as well.el

**Time Spent Reviewing:**

3 hours

---

> ### Author Response · Authors · 2021-08-10
> **Response to Reviewer 5Lqk**
>
> Q. “Is there a reason why the algorithm of Lee et al. (2021) would not extend to the setting studied in this paper?”
>
> A. In the setting of Lee et al. (2021), the corrupted reward should be linear in the actions. As a result, their work can not be extended to our setting in a straightforward way. (For example, eq (20) in their proof does not hold in our setting. )
>
> Q. It seems that the paper does not allow the variance to also be corrupted and it also requires $\sigma_t$ to be observable. It would be great if these assumptions are necessary or if they are for the convenience of exposition.  A related question: Do we know anything about the case where $\sigma_t$  is not observable in the uncorrupted case?
>
> A. The assumption that the variance $\sigma_t$ is known is only made for ease of analysis. Otherwise, we need to know an upper bound on $\sigma_t$. On the other hand, our current analysis does not allow the variance to also be corrupted. This is an interesting setting and we will think more about it. We are aware of a work [arXiv:2101.12745] for the case where $\sigma_t$ is not observable in the uncorrupted case, yet the algorithm seems to be computationally inefficient.
>
> Q. It would be useful if the paper pinpoints what is the feature that exists in linear contextual bandits that requires moving beyond elimination-based techniques.
>
> A. Thank you for pointing out the interesting papers. We will rephrase our comment on the arm elimination algorithm to make it more accurate.
>
> Q. “It would be great if the authors could explain whether there is some roadblock that arises in the intersection of these techniques that does not exist in each of them independently.”
>
> A. Actually either the technique of subsampling or the design of multiple levels of confidence set cannot provide a regret upper bound alone following our analysis. Thus we believe the main roadblock lies in how to combine the techniques to achieve the regret bound.

---

### Decision · Program_Chairs · 2021-09-27

**Decision:**

Reject

**Comment:**

While the review team appreciated the model and the new algorithms designed, they also identified a couple of weaknesses which I summarize below:

* While the results are interesting, they follow from a combination of previously known techniques.
* Dependency $C^2$ seems suboptimal and un-natural
* There is no strong evidence of why a simpler algorithm (like Corral or MWU on the corruption level) wouldn't work here.
* The model could be better motivated (e.g. why is the variance revealed by the adversary).

Although the reviewers address some of that in their response, that wasn't enough to substantially change the sentiment towards the paper.